# Multi-layer random features and the approximation power of neural networks

**Rustem Takhanov**[1]

[1]Mathematics Dept., Nazarbayev University, Astana, Kazakhstan

## Abstract

A neural architecture with randomly initialized weights, in the infinite width limit, is equivalent to a Gaussian Random Field whose covariance function is the so-called Neural Network Gaussian Process kernel (NNGP). We prove that a reproducing kernel Hilbert space (RKHS) defined by the NNGP contains only functions that can be approximated by the architecture. To achieve a certain approximation error the required number of neurons in each layer is defined by the RKHS norm of the target function. Moreover, the approximation can be constructed from a supervised dataset by a random multi-layer representation of an input vector, together with training of the last layer's weights. For a 2-layer NN and a domain equal to an $n-1$-dimensional sphere in $\mathbb{R}^n$, we compare the number of neurons required by Barron's theorem and by the multi-layer features construction. We show that if eigenvalues of the integral operator of the NNGP decay slower than $k^{-n-\frac{2}{3}}$ where $k$ is an order of an eigenvalue, then our theorem guarantees a more succinct neural network approximation than Barron's theorem. We also make some computational experiments to verify our theoretical findings. Our experiments show that realistic neural networks easily learn target functions even when both theorems do not give any guarantees.

## 1 INTRODUCTION

Kernel methods in machine learning (ML) is a classical research topic that has found applications in classification/regression Steinwart and Christmann [2008], dimension reduction Fukumizu et al. [2009], generative modeling Li et al. [2015], probability density estimation, non-parametric statistics, spline interpolation Wahba [1990], and many other areas. Being a fundamental mathematical object, kernels are not only applicable in practice but also suitable for theoretical analysis. Recently the field became quite active again due to the discovered fact that neural networks (NN), under the so-called infinite width limit, behave pretty much like the kernel regression. To any architecture of a neural network one can correspond a specific kernel, called the neural tangent kernel (NTK) Jacot et al. [2018], whose structure is defined by the geometry of a reproducing kernel Hilbert space (RKHS). A major question in this field is to identify aspects of gradient-based learning with this architecture that can be explained by the NTK.

The NTK is not the first kernel that appeared in the theory of neural networks. Another interesting case is the Neural Network Gaussian Process kernel (NNGP), which was suggested earlier by Neal [1996]. Unlike the NTK, the NNGP does not explain the behavior of NNs trained by gradient descent, but it helps to understand the structure of an NN whose weights are initialized randomly. It turns out that when a distribution of weights is normal (with zero mean and proper scaling of the variance), in the infinite width limit, an NN behaves like a Gaussian Random Field whose covariance function is the NNGP. Moreover, as was shown by Daniely et al. [2016], random networks induce representations that approximate the RKHS defined by the NNGP.

The mentioned results motivate us to formulate the following question: is a ball in the RKHS of the NNGP a natural set of functions approximated well by a given architecture of NNs? To answer the question, we consider a feed-forward NN architecture with a non-linearity $\sigma : \mathbb{R} \to \mathbb{R}$, an architecture with $L$ hidden layers of neurons and a one-dimensional output, i.e. the mapping $\mathbf{x} \to \mathbf{w}^\top \sigma(W^{(L)} \sigma(\cdots \sigma(W^{(1)}\mathbf{x}) \cdots))$ parameterized by matrices $W^{(h)} \in \mathbb{R}^{n_h \times n_{h-1}}$, $\mathbf{w} \in \mathbb{R}^{n_L}$ (we will call them $L+1$-NN). In the infinite width limit, such an architecture is fully defined by $n$ and $\sigma$ itself.

Based on our understanding of the RKHS of the NNGP $K$, denoted by $\mathcal{H}_K$, as a space that is "native" to the NN

architecture, we expect that a statement similar to Barron's theorem should hold, in which the complexity of a function is measured by $\|f\|_{\mathcal{H}_K}$, instead of the Barron norm, denoted by $C_{f,\boldsymbol{\Omega}}$. Indeed, we prove a general statement (Theorem 4) whose specification for $L = 1$ looks very analogous to Barron's theorem, with a role of $\|f\|_{\mathcal{H}_K}$ being analogous to the role of $C_{f,\boldsymbol{\Omega}}$.

Theorem 4 guarantees that the unit ball in $\mathcal{H}_K$, denoted by $B_{\mathcal{H}_K}$, indeed contains only functions that are very well approximable by our architecture. For $L = 1$ we only require that the activation function $\sigma$ is bounded. For many practical activation functions, this condition is satisfied. For the ReLU it is not satisfied, but it is satisfied for $\sigma_1(x) = \mathrm{ReLU}(x) - \mathrm{ReLU}(x - 1)$ and therefore, the number of neurons required to approximate a function $f$ by a ReLU 2-NN is proportional to $\|f\|_{\mathcal{H}_K}^2$ where $K$ is the NNGP for $\sigma_1$. The multi-layer case ($L \geq 2$) requires boundedness of all derivatives up to the fourth degree, which is satisfied for such activation functions as a sigmoid, a hyperbolic tangent, erf, a cosine, and a Gaussian.

To put our findings into a broader context of approximation theory, we question whether an approximation guarantee of Theorem 4 for $L = 1$ gives any advantage over the classical Barron's theorem. This poses a general problem: how are the Barron space for $\boldsymbol{\Omega}$ and the RKHS $\mathcal{H}_K$ related? Specifically, can we say that some functions for which Theorem 4 guarantees the existence of a succinct representation in our architecture, require too many neurons according to Barron's theorem? In other words, which activation functions have an unbounded (or bounded) set $B_{\mathcal{H}_K}$ w.r.t. the norm $C_{f,\boldsymbol{\Omega}}$. We address this problem in the paper and characterize activation functions for which $B_{\mathcal{H}_K}$ is an unbounded/bounded set w.r.t. the Barron norm for $\boldsymbol{\Omega} = \mathbb{S}^{n-1}$.

**Related work.** Besides the mentioned works, the topic of the approximation power of NNs attracted a lot of attention. The approximation power of 2-NNs was a topic of classical works Cybenko [1989], Hornik [1991], Leshno et al. [1993], with a key result being Barron's theorem Barron [1993]. A certain generalization of Barron's theorem to multi-layer networks is presented in Lee et al. [2017]. An approach to NN training based on random nonlinear features was introduced in Rahimi and Recht [2008] and further developed in Daniely et al. [2017], Bach [2017b]. Improvements in an approximation ability of NNs from increasing depth were demonstrated in Telgarsky [2015], Eldan and Shamir [2016]. Similarities between behaviors of randomly initialized multi-layer NNs in the infinite width limit and Gaussian Processes are discussed in Williams [1996], Lee et al. [2018], Bracale et al. [2021], Zhang et al. [2024].

## 2 PRELIMINARIES AND NOTATIONS

Bold-faced lowercase letters ($\mathbf{x}$) denote (random) vectors, and regular lowercase letters ($x$) denote scalars. $\|\cdot\|$ denotes the Euclidean norm: $\|\mathbf{x}\| := \sqrt{\mathbf{x}^\top \mathbf{x}}$. For any distribution $\mathcal{P}$, sampling $\mathbf{x}$ from $\mathcal{P}$ is denoted by $\mathbf{x} \sim \mathcal{P}$. Given a Borel set $\boldsymbol{\Omega}$ and a Borel measure $\mu$ on $\boldsymbol{\Omega}$, by $L_2(\boldsymbol{\Omega}, \mu)$ we denote the completion of $\mathcal{H}_0$, where $\mathcal{H}_0$ is a space of real-valued functions on $\boldsymbol{\Omega}$ with the inner product $\langle u, v \rangle_{\mathcal{H}_0} = \int_{\boldsymbol{\Omega}} u(\mathbf{x}) v(\mathbf{x}) d\mu(\mathbf{x})$. The corresponding inner product is denoted by $\langle \cdot, \cdot \rangle_{L_2(\boldsymbol{\Omega}, \mu)}$ and the induced norm is then $\|u\|_{L_2(\boldsymbol{\Omega}, \mu)} = \sqrt{\langle u, u \rangle_{L_2(\boldsymbol{\Omega}, \mu)}}$. If $d\mu = p(\mathbf{x}) d\mathbf{x}$, then $L_2(\boldsymbol{\Omega}, \mu)$ is denoted by $L_2(\boldsymbol{\Omega}, p)$. Analogously, the Banach space $L_p(\boldsymbol{\Omega}, \mu)$ is defined. Given a Mercer kernel $K : \boldsymbol{\Omega} \times \boldsymbol{\Omega}$, $\mathcal{H}_K$ denotes a reproducing kernel Hilbert space defined by $K$. Then, $B_{\mathcal{H}_K}$ denotes a unit ball centered at $\mathbf{0}$ in the RKHS $\mathcal{H}_K$. The Fourier transform of a function $a : \mathbb{R}^n \to \mathbb{C}$ is denoted by $\widehat{a}$.

Given $f : \mathbb{R} \to \mathbb{R}$ and $g : \mathbb{R} \to \mathbb{R}_+$, we write $f \ll g$ if there exist universal constants $\alpha, \beta \in \mathbb{R}_+$ such that for all $x > \beta$ we have $|f(x)| \leq \alpha g(x)$. When $f, g : \mathbb{R} \to \mathbb{R}_+$, we write $f \asymp g$ if $f \ll g$ and $g \ll f$. If in an equation we are not interested in a factor depending on the dimension $n$ we write $f \propto^n g$, and that means $f = c_n g$ for some constant $c_n$. Analogously, $f \ll^n g$ means $f \ll c_n g$.

Proofs of all given statements can be found in the Appendix.

## 3 FULLY CONNECTED FEED-FORWARD NEURAL NETWORK AND ASSOCIATED KERNELS

Let $\mathbf{x} \in \mathbb{R}^{n_0}$ be the input and $n_1, \cdots, n_L$ be dimensions of hidden layers. We denote $\theta = [W^{(1)}, \cdots, W^{(L)}]$ where $W^{(h)} \in \mathbb{R}^{n_h \times n_{h-1}}$. Let us denote $\alpha^{(0)}(\mathbf{x}, \theta) = \mathbf{x}$ and

$$\tilde{\alpha}^{(h)}(\mathbf{x}, \theta) = W^{(h)} \alpha^{(h-1)}(\mathbf{x}, \theta),$$
$$\alpha^{(h)}(\mathbf{x}, \theta) = \sigma(\tilde{\alpha}^{(h)}(\mathbf{x}, \theta)), h = 1, \cdots, L.$$

Then, $\tilde{\alpha}^{(h)} = [\tilde{\alpha}_i^{(h)}]_{i=1}^{n_h}$ are called preactivations and $\alpha^{(h)} = [\alpha_i^{(h)}]_{i=1}^{n_h}$ are called activations. If we sample entries of $W^{(h)} = [W_{ij}^{(h)}]$ independently according to $W_{ij}^{(1)} \sim \mathcal{N}(0, 1)$ and $W_{ij}^{(h)} \sim \mathcal{N}(0, \frac{1}{n_{h-1}}), h = 2, \cdots, L$, then sending $n_1, \cdots, n_{L-1} \to +\infty$ makes $\{\tilde{\alpha}_i^{(h+1)}(\mathbf{x}, \theta)\}$ the Gaussian Random Field (for any $i \in [n_{h+1}]$) with the covariance function

$$\mathbb{E}[\tilde{\alpha}_i^{(h+1)}(\mathbf{x}, \theta) \tilde{\alpha}_i^{(h+1)}(\mathbf{x}', \theta)] \to \Sigma^{(h)}(\mathbf{x}, \mathbf{x}'),$$

where the kernels $\Sigma^{(h)}, h = 1, \cdots, L$, called Neural Network Gaussian Process (NNGP) kernels, are defined according to

$$\Sigma^{(0)}(\mathbf{x}, \mathbf{x}') = \mathbf{x}^\top \mathbf{x}',$$
$$\Sigma^{(h+1)}(\mathbf{x}, \mathbf{x}') = \mathbb{E}_{(u,v) \sim \Lambda^{(h)}(\mathbf{x}, \mathbf{x}')}[\sigma(u)\sigma(v)], \quad (1)$$

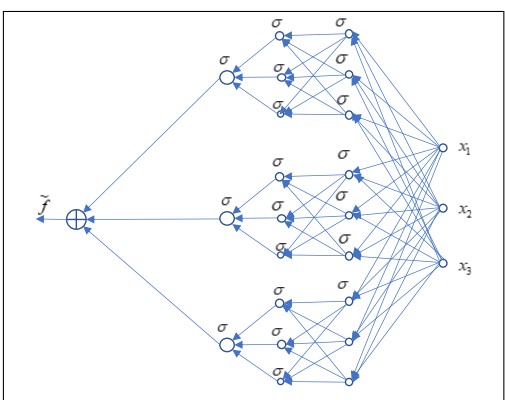

Figure 1: An architecture for $n_0 = 3, n_1 = n_2 = 3, n_3 = 1, T = 3$.

where $\Lambda^{(h)}(\mathbf{x}, \mathbf{x}') = \begin{bmatrix} \Sigma^{(h)}(\mathbf{x}, \mathbf{x}) & \Sigma^{(h)}(\mathbf{x}, \mathbf{x}') \\ \Sigma^{(h)}(\mathbf{x}, \mathbf{x}') & \Sigma^{(h)}(\mathbf{x}', \mathbf{x}') \end{bmatrix}$.

In the regime of finite $n_1, \cdots, n_{L-1}$, we introduce kernels $\tilde{\Sigma}^{(h)}, h = 1, \cdots, L$ that approximate kernels of the infinite-width limit, i.e.

$$\tilde{\Sigma}^{(h)}(\mathbf{x}, \mathbf{x}') = \mathbb{E}_{W_1, \cdots, W_h}[\alpha_i^{(h)}(\mathbf{x}, \theta) \alpha_i^{(h)}(\mathbf{x}', \theta)]. \quad (2)$$

Since all entries of $\alpha^{(h)}(\mathbf{x}, \theta)$ have the same distribution, the latter expression is the same for any $i \in [n_h]$. It is also natural to approximate $\tilde{\Sigma}^{(h)}$ by its empirical version, i.e. by

$$\Sigma_{\text{emp}}^{(h)}(\mathbf{x}, \mathbf{x}') = \frac{1}{n_h} \sum_{i=1}^{n_h} \alpha_i^{(h)}(\mathbf{x}, \theta) \alpha_i^{(h)}(\mathbf{x}', \theta). \quad (3)$$

By construction, we have $\mathbb{E}_{W_1, \cdots, W_h}[\Sigma_{\text{emp}}^{(h)}(\mathbf{x}, \mathbf{x}')] = \tilde{\Sigma}^{(h)}(\mathbf{x}, \mathbf{x}')$ and $\lim_{n_h \to +\infty} \cdots \lim_{n_1 \to \infty} \tilde{\Sigma}^{(h)}(\mathbf{x}, \mathbf{x}') = \Sigma^{(h)}(\mathbf{x}, \mathbf{x})$. By analogy, we define $\tilde{\Lambda}^{(h)}(\mathbf{x}, \mathbf{x}') = \begin{bmatrix} \tilde{\Sigma}^{(h)}(\mathbf{x}, \mathbf{x}) & \tilde{\Sigma}^{(h)}(\mathbf{x}, \mathbf{x}') \\ \tilde{\Sigma}^{(h)}(\mathbf{x}, \mathbf{x}') & \tilde{\Sigma}^{(h)}(\mathbf{x}', \mathbf{x}') \end{bmatrix}$ and $\Lambda_{\text{emp}}^{(h)}(\mathbf{x}, \mathbf{x}') = \begin{bmatrix} \Sigma_{\text{emp}}^{(h)}(\mathbf{x}, \mathbf{x}) & \Sigma_{\text{emp}}^{(h)}(\mathbf{x}, \mathbf{x}') \\ \Sigma_{\text{emp}}^{(h)}(\mathbf{x}, \mathbf{x}') & \Sigma_{\text{emp}}^{(h)}(\mathbf{x}', \mathbf{x}') \end{bmatrix}$.

## 4  MAIN RESULTS

A starting point of our approach to approximate functions by multi-layer NNs is the following remarkable property of the finite version of the NNGP kernel, $\tilde{\Sigma}^{(L)}$.

**Theorem 1.** *Let $\mu$ be a probabilistic measure on $\boldsymbol{\Omega} \subseteq \mathbb{R}^n$, $\sigma$ be bounded, and $n_1, \cdots, n_L, T \in \mathbb{N}$, $n_L = 1$. Then, for any $f \in \mathcal{H}_{\tilde{\Sigma}^{(L)}}$ there exist matrices $W^{(i,h)} \in \mathbb{R}^{n_h \times n_{h-1}}$, where $h = 1, \cdots, L$, $i = 1, \cdots, T$, and weights $w_i \in \mathbb{R}, i = 1, \cdots, T$ such that*

$$\|f(\mathbf{x}) - \tilde{f}(\mathbf{x})\|_{L_2(\boldsymbol{\Omega}, \mu)} \leq \frac{\|\sigma\|_\infty \|f\|_{\mathcal{H}_{\tilde{\Sigma}^{(L)}}}}{\sqrt{T}}.$$

*where $\tilde{f}(\mathbf{x}) = \sum_{i=1}^T w_i \sigma(W^{(i,L)} \sigma(\cdots \sigma(W^{(i,1)} \mathbf{x}) \cdots))$.*

A proof of the latter statement is based on the representation (2) of the kernel $\tilde{\Sigma}^{(L)}$ as an inner product between functions $\alpha^{(L)}(\mathbf{x}, \cdot)$ and $\alpha^{(L)}(\mathbf{x}', \cdot)$ in the corresponding space and is in line with some earlier results obtained for other classes of functions (see Proposition 4.1 from Rahimi and Recht [2008] or Corollary 4 from Daniely et al. [2017]). The approximating function $\tilde{f}$ can be viewed as a feed-forward neural network with $L + 1$ layers whose neurons of the first $L$ layers are divided into $T$ parts of equal size that are connected by a final $L + 1$-st layer (an example of that architecture is shown in Figure 1). From arguments of the proof it is clear that $\tilde{f}$ has the following structure: all matrices $W^{(i,h)}$, $h = 1, \cdots, L$, $i = 1, \cdots, T$ are sampled independently according to $W_{kl}^{(i,1)} \sim \mathcal{N}(0, 1)$, $W_{kl}^{(i,h)} \sim \mathcal{N}(0, \frac{1}{n_{h-1}})$, $h = 2, \cdots, L$; afterward, we set $w_i = \frac{1}{T} g(W^{(i,1)}, \cdots, W^{(i,L)})$ where $g$ is some function. That is, the last layer's weights are defined by the previous layer's random initialization. In practice, the weight vector $\mathbf{w} = [w_i]_{i=1}^T$ can be computed from a supervised dataset $\{(\mathbf{x}_s, f(\mathbf{x}_s))\}_{s=1}^N, \{\mathbf{x}_i\} \sim^{\text{iid}} \mu$ by a standard linear regression formula $\mathbf{w} = (X^\top X)^{-1} X^\top [f(\mathbf{x}_s)]_{s=1}^N$ where $X \in \mathbb{R}^{N \times n_L}$ is a design matrix whose $s$th row is a representation of $\mathbf{x}_s$ by $T$ activations $\{\sigma(W^{(i,L)} \sigma(\cdots \sigma(W^{(i,1)} \mathbf{x}_s) \cdots))\}_{i=1}^T$. It is natural to call this approach to construct the approximating function $\tilde{f}$ *a multi-layer random feature model (ML-RFM).*

$\tilde{\Sigma}^{(L)}$, unlike the NNGP kernel $\Sigma^{(L)}$, is hard to analyze both analytically and numerically. So, our first goal was to study the cost of substituting the empirical kernel $\tilde{\Sigma}_{\text{emp}}^{(L)}$ or the NNGP kernel $\Sigma^{(L)}$ for $\tilde{\Sigma}^{(L)}$. The empirical kernel $\tilde{\Sigma}_{\text{emp}}^{(L)}$ is a random variable whose mean is $\tilde{\Sigma}^{(L)}$, and a variance of it is a natural measure of the distance between them. As the following theorem demonstrates, each layer contributes to this variance a term inverse proportional to the layer's size.

**Theorem 2.** *For bounded $\sigma, \sigma', \sigma''$ and $h = 1, \cdots, L$, we have*

$$\text{Var}[\Sigma_{\text{emp}}^{(h)}(\mathbf{x}, \mathbf{x}')] \leq 2\|\sigma\|_\infty^4 \sum_{i=1}^h \frac{C^{h-i}}{n_i},$$

*where $C = 4\max(\|\sigma''\|_\infty \|\sigma\|_\infty, \|\sigma'\|_\infty^2)^2$.*

A proof of the latter theorem is based on the following observation. From the law of total variance it is clear that $\text{Var}[\Sigma_{\text{emp}}^{(h)}(\mathbf{x}, \mathbf{x}')]$ consists of two parts: the first part is an expectation of $\text{Var}[\Sigma_{\text{emp}}^{(h)}(\mathbf{x}, \mathbf{x}') \mid W_1, \cdots, W_{h-1}]$, and the second part is the variance of $\mathbb{E}[\Sigma_{\text{emp}}^{(h)}(\mathbf{x}, \mathbf{x}') \mid W_1, \cdots, W_{h-1}]$. From the definition (3) it can be seen that the first part behaves like $\mathcal{O}(\frac{1}{n_h})$ as $\Sigma_{\text{emp}}^{(h)}(\mathbf{x}, \mathbf{x}')$ is an average of $n_h$ independent terms (given $W_1, \cdots, W_{h-1}$). We show that the second term is bounded by a combination of variances of $\Sigma_{\text{emp}}^{(h-1)}(\mathbf{x}, \mathbf{x}')$, $\Sigma_{\text{emp}}^{(h-1)}(\mathbf{x}, \mathbf{x})$, and $\Sigma_{\text{emp}}^{(h-1)}(\mathbf{x}', \mathbf{x}')$. This allows us to bound the needed variance by variances of empirical kernels of lower layers. Applying this argument iteratively leads us to the bound of Theorem 2.

Being interesting in itself, the previous theorem is instrumental in proving the following estimate of the difference between the finite version of the NNGP, $\tilde{\Sigma}^{(L)}$, and the NNGP.

**Theorem 3.** *Let $\sigma$ be such that $\sigma, \sigma', \sigma'', \sigma''', \sigma''''$ are bounded and continuous. Then, there exists a universal constant $R$ such that*

$$|\tilde{\Sigma}^{(L)}(\mathbf{x}, \mathbf{x}') - \Sigma^{(L)}(\mathbf{x}, \mathbf{x}')| \leq$$

$$R\|\sigma\|_\infty^4 \max(\|\sigma''''\|_\infty\|\sigma\|_\infty, \|\sigma'''\|_\infty\|\sigma'\|_\infty, \|\sigma''\|_\infty^2)$$

$$\sum_{j=1}^{L-1} \frac{\max(2\|\sigma''\|_\infty\|\sigma\|_\infty, 2\|\sigma'\|_\infty^2, \frac{5}{2})^{2L-2j}(L-j)}{n_j}.$$

If the RHS of the inequality from Theorem 3 is small, then it is natural to expect that two spaces, $\mathcal{H}_{\tilde{\Sigma}^{(L)}}$ and $\mathcal{H}_{\Sigma^{(L)}}$, approximate each other. This allows to translate the desirable property of $\mathcal{H}_{\tilde{\Sigma}^{(L)}}$ from Theorem 1 to the space $\mathcal{H}_{\Sigma^{(L)}}$.

Further, we assume that $\Omega \subseteq \mathbb{R}^n$ is compact. A Borel measure $\mu$ on $\Omega$ is called nondegenerate if for any open set $S \subseteq \mathbb{R}^n$ such that $S \cap \Omega \neq \emptyset$, we have $\mu(S \cap \Omega) \neq 0$.

**Theorem 4.** *Let $\mu$ be a probabilistic nondegenerate Borel measure on compact $\Omega \subseteq \mathbb{R}^n$ and $\sigma$ be such that $\sigma, \sigma', \sigma'', \sigma''', \sigma''''$ are bounded and continuous. We also assume that $n_1, \cdots, n_L, T \in \mathbb{N}$, $n_L = 1$. Then, for any $f \in \mathcal{H}_{\Sigma^{(L)}}$ there exist matrices $W^{(i,h)} \in \mathbb{R}^{n_h \times n_{h-1}}$, where $h = 1, \cdots, L$, $i = 1, \cdots, T$, and weights $w_i \in \mathbb{R}, i = 1, \cdots, T$ such that*

$$\|f(\mathbf{x}) - \tilde{f}(\mathbf{x})\|_{L_2(\Omega, \mu)} \leq \|f\|_{\mathcal{H}_{\Sigma^{(L)}}} \Big( \frac{\|\sigma\|_\infty}{\sqrt{T}} +$$

$$cC_1 \Big( \sum_{j=1}^{L-1} \frac{C_2^{2L-2j}(L-j)}{n_j} \Big)^{1/2} \Big).$$

*where $\tilde{f}(\mathbf{x}) = \sum_{i=1}^T w_i \sigma(W^{(i,L)}\sigma(\cdots \sigma(W^{(i,1)}\mathbf{x}) \cdots))$, $c$ is a universal constant and*

$$C_1 = \|\sigma\|_\infty^2 \sqrt{\max(\|\sigma''''\|_\infty\|\sigma\|_\infty, \|\sigma'''\|_\infty\|\sigma'\|_\infty, \|\sigma''\|_\infty^2)},$$

$$C_2 = \max(2\|\sigma''\|_\infty\|\sigma\|_\infty, 2\|\sigma'\|_\infty^2, \frac{5}{2}).$$

**Remark 1.** *If $L = 1$, then the second term in the RHS of the latter inequality is absent. It can be seen from the proof of Theorem 4 that this case requires only that $\sigma$ is bounded. Theorem says that for any $f \in \mathcal{H}_{\Sigma^{(1)}}$ and $T \in \mathbb{N}$ there exist $\mathbf{a}_1, \cdots, \mathbf{a}_T \in \mathbb{R}^n, b_1, \cdots, b_T \in \mathbb{R}$ such that*

$$\|f - \sum_{i=1}^T b_i \sigma(\mathbf{a}_i^\top \mathbf{x})\|_{L_2(\Omega, \mu)} \leq \frac{\|\sigma\|_\infty\|f\|_{\mathcal{H}_{\Sigma^{(1)}}}}{\sqrt{T}}. \quad (4)$$

**Remark 2.** *If we assume that all derivatives of $\sigma$ up to fourth degree and $L$ are bounded by some universal constant, then we have*

$$\|f(\mathbf{x}) - \tilde{f}(\mathbf{x})\|_{L_2(\Omega, \mu)} \ll \frac{\|f\|_{\mathcal{H}_{\Sigma^{(L)}}}}{\sqrt{\min(T, n_1, \cdots, n_{L-1})}}.$$

*Thus, to achieve $\|f(\mathbf{x}) - \tilde{f}(\mathbf{x})\|_{L_2(\Omega, \mu)} = \mathcal{O}(\varepsilon)$ we need*

$$\min(T, n_1, \cdots, n_{L-1}) = \mathcal{O}\big(\frac{\|f\|_{\mathcal{H}_{\Sigma^{(L)}}}^2}{\varepsilon^2}\big).$$

## 4.1 A RELATIONSHIP WITH THE BARRON SPACE

Let us consider the case of $L = 1$. There is a direct analogy between the inequality (4) and Barron's theorem. To demonstrate that, let us introduce the Barron norm using a recent exposition from Lee et al. [2017].

**Definition 1.** *For a bounded set $\Omega \subseteq \mathbb{R}^n$ let us define $\|\boldsymbol{\omega}\|_\Omega = \sup_{\mathbf{x} \in \Omega} |\boldsymbol{\omega}^\top \mathbf{x}|$. Let $\mathcal{F}_\Omega$ be a set of functions $g : \mathbb{R}^n \to \mathbb{R}$ with existing Fourier transform $\widehat{g}$ such that*

$$\forall \mathbf{x} \in \Omega, \ g(\mathbf{x}) - g(\mathbf{0}) = \int_{\mathbb{R}^n} (e^{i\boldsymbol{\omega}^\top \mathbf{x}} - 1)\widehat{g}(\boldsymbol{\omega})d\boldsymbol{\omega}.$$

*Then, for a function $f : \Omega \to \mathbb{R}$ we define its $\Omega$-norm as*

$$C_{f,\Omega} = \inf_{g \in \mathcal{F}_\Omega: g|_\Omega = f} \int_{\mathbb{R}^n} \|\boldsymbol{\omega}\|_\Omega |\widehat{g}(\boldsymbol{\omega})|d\boldsymbol{\omega}. \quad (5)$$

*With a slight abuse of terminology we call the set of functions with a finite $\Omega$-norm the Barron space of $\Omega$.*

Since the infimum in (5) is taken over all possible extensions of $f$, even an approximate computation of it is a non-trivial problem. Barron's theorem claims that any function $f$ from the Barron space of $\Omega$ can be approximated by a two-layer neural network $\tilde{f}(\mathbf{x}) = \sum_{i=1}^T b_i \sigma(\mathbf{a}_i^\top \mathbf{x} + c_i)$ in such a way that $\|f - \tilde{f}\|_{L_2(\Omega, \mu)} \ll \frac{C_{f,\Omega}}{\sqrt{T}}$. The norm $C_{f,\Omega}$ in Barron's theorem plays the role of the function's complexity and is analogous to $\|f\|_{\mathcal{H}_{\Sigma^{(1)}}}$ in (4).

This subsection is dedicated to describing a relationship between these two norms, $C_{f,\Omega}$ and $\|f\|_{\mathcal{H}_{\Sigma^{(1)}}}$, for a special domain $\Omega = \mathbb{S}^{n-1}$, where $\mathbb{S}^{n-1}$ is a sphere of unit radius in $\mathbb{R}^n$. The case $\Omega = \mathbb{S}^{n-1}$ plays a special role in the analysis of NNGPs Bach [2017a], Geifman et al. [2020], Chen and Xu [2021] due to the fact that $\Sigma^{(L)}(\mathbf{x}, \mathbf{y}) = k(\mathbf{x}^\top \mathbf{y})$ for some function $k$, i.e. the NNGP $\Sigma^{(L)}$ is the so-called zonal kernel. We analyze this issue to find the conditions under which an approximation error guaranteed by the random features model (RFM) is better than an approximation error guaranteed by Barron's theorem. Since our results hold for any zonal kernel (not necessarily the NNGP kernel), we will formulate them for a general zonal kernel $K$.

A well-known fact from the theory of RKHSs states that $\mathcal{H}_K$ is isomorphic to $\mathcal{L} = O_K^{1/2}[L_2(\Omega, \nu)]$ equipped with the inner product $\langle O_K^{1/2}[f], O_K^{1/2}[g] \rangle_\mathcal{L} = \langle f, g \rangle_{L_2(\Omega, \nu)}$ where $O_K : L_2(\Omega, \nu) \to L_2(\Omega, \nu)$ is defined by $O_K[f](\mathbf{x}) = \int_\Omega K(\mathbf{x}, \mathbf{y})f(\mathbf{y})d\nu(\mathbf{y})$ and $\nu$ is assumed to be non-degenerate on $\Omega$ Cucker and Zhou [2007].

A measure $\nu$ can be defined as the surface volume measure on $\mathbb{S}^{n-1}$ and eigenvectors of $O_K$ are real-valued spherical

harmonics. Let $Y_{k,j} : \mathbb{S}^{n-1} \to \mathbb{R}, j = 1, \cdots, N(n,k)$ be an orthonormal basis in a space of spherical harmonics of order $k = 0, 1, \cdots$ (w.r.t. the inner product in $L_2(\mathbb{S}^{n-1}, \nu)$). Then for any $\mathbf{x}, \mathbf{y} \in \mathbb{S}^{n-1}$ we have

$$K(\mathbf{x}, \mathbf{y}) = \sum_{k=0}^{\infty} \lambda_k \sum_{j=1}^{N(n,k)} Y_{k,j}(\mathbf{x}) Y_{k,j}(\mathbf{y}),$$

where $O_K[Y_{k,j}] = \lambda_k Y_{k,j}$, i.e. $\lambda_k$ is an eigenvalue of $O_K$. For more information on spherical harmonics, we refer to Frye and Efthimiou [2012].

Thus, the RKHS $\mathcal{H}_K$ can be characterized as the set

$$\Big\{ \sum_{k=0}^{\infty} \sigma_k \sum_{j=1}^{N(n,k)} x_{kj} Y_{k,j} \mid \sum_{k=0}^{\infty} \sum_{j=1}^{N(n,k)} x_{kj}^2 < \infty \Big\}.$$

where $\sigma_k = \sqrt{\lambda_k}$.

Our first result claims that if eigenvalues $\{\lambda_k\}$ decay slowly enough, then $B_{\mathcal{H}_K}$ is an unbounded set w.r.t. the norm $C_{f,\mathbb{S}^{n-1}}$.

**Theorem 5.** *Let $K$ be a zonal Mercer kernel and $\{\lambda_k\}$ be its eigenvalues. If $\limsup\limits_{k \to +\infty} \frac{\lambda_k k^{n+\frac{2}{3}}}{\sqrt{\log k}} = +\infty$, then $B_{\mathcal{H}_K}$ is an unbounded set in the Barron space of $\mathbb{S}^{n-1}$.*

This result can be directly applied to almost all popular activation functions. E.g., for the function $\sigma(x) = x_+^{\alpha}$ where $x_+ = \frac{x+|x|}{2}$, eigenvalues of the NNGP for a neural network with a single hidden layer were calculated in Bach [2017a]. It was shown that $\lambda_k \asymp c_n k^{-n}$ if $k > 0$ is even. This case captures the step function ($\alpha = 0$) and the ReLU activation function ($\alpha = 1$). The previous theorem implies that $\frac{C_{f,\mathbb{S}^{n-1}}}{\|f\|_{\mathcal{H}_K}}$ can be made arbitrarily large. If an activation function is bounded additionally, e.g. as the step function, then according to Remark 1, some functions have a succinct representation as a 2-Layer NNs (that can be found using RFM), with a much better approximation error than the one guaranteed by Barron's theorem.

A corresponding inclusion result is given below.

**Theorem 6.** *Let $K$ be a zonal Mercer kernel and $\{\lambda_k\}$ be its eigenvalues. If $\sum_{k=0}^{\infty} \lambda_k k^{n+\frac{2}{3}} < +\infty$, then $B_{\mathcal{H}_K}$ is a bounded set in the Barron space of $\mathbb{S}^{n-1}$.*

Thus, if eigenvalues decay substantially faster than $\frac{1}{k^{n+\frac{2}{3}}}$, e.g. exponentially fast, one can derive that $C_{f,\mathbb{S}^{n-1}} \leq c\|f\|_{\mathcal{H}_K}$ for some constant $c > 0$. This is the case when the representation guaranteed by Theorem 4 is not shorter than the representation of Barron's theorem. Examples of activation functions for which eigenvalues decay very fast include a) the Gaussian function, b) the cosine function, and c) the sine function. Indeed, the following theorems hold (their proofs can be found in the Appendix H and the Appendix I).

**Theorem 7.** *Let $\sigma(x) = e^{-\frac{x^2}{2}}$, $\Omega = \mathbb{S}^{n-1}$ and $K$ is the NNGP kernel given by (1). Then, $\lambda_{2k+1} = 0$ and $\lambda_{2k} \ll^n 2^{-2k} k^{-\frac{n}{2}}$.*

**Theorem 8.** *Let $\Omega = \mathbb{S}^{n-1}$ and $K$ be defined by (1). For the case $\sigma(x) = \cos(ax)$, we have $\lambda_{2k+1} = 0$ and $\lambda_{2k} \ll^n \frac{a^{4k}}{\sqrt{k} 2^{2k} \Gamma(2k+\frac{n-1}{2})}$. Analogously, for the case $\sigma(x) = \sin(ax)$, we have $\lambda_{2k} = 0$ and $\lambda_{2k+1} \ll^n \frac{a^{4k+2}}{\sqrt{k} 2^{2k} \Gamma(2k+\frac{n+1}{2})}$.*

To summarize, we demonstrate that there is a sharp difference between two types of activation functions, those for which eigenvalues of $O_K$ decay slower than $k^{-n-\frac{2}{3}}$ (modulo a logarithmic factor) and those for which eigenvalues decay much faster than $k^{-n-\frac{2}{3}}$. For the first type of activation functions, we can guarantee that 2-NNs trained by RFM can approximate functions that are not captured by Barron's theorem.

**Remark 3.** *In the proof of Theorem 5 we construct a function $Y_k$ (its structure is described in Lemma 9) that belongs to the space of harmonics of order $k$ and has a unit $L_2(\mathbb{S}^{n-1})$-norm as well as a moderate $L_\infty(\mathbb{S}^{n-1})$-norm. Our analysis shows that norms of that function in the Barron space of $\mathbb{S}^{n-1}$ and in $\mathcal{H}_{\Sigma^{(1)}}$ (for all popular activation functions $\sigma$) rapidly grow with an increase of $k$ and blow up for moderate $k$. In the experimental part of the paper (Section 5) we study the learnability of this function using 2-NNs by RFM and a gradient-based algorithm. Our results show that $Y_k$ is not a hard target for a gradient-based algorithm even for moderate $k$'s. We discuss that this example shows that the approximation power of 2-NNs, as well as their learnability by the gradient descent, are definitely beyond the guarantees of Barron's theorem, the RFM, and the NTK theory.*

## 5 EXPERIMENTS

**Decay rate of eigenvalues for popular activation functions.** As pointed out in Section 4.1, activation functions can be conventionally classified into two classes: those for which Theorem 4 guarantees the existence of functions which has (a) a large norm in the Barron space and (b) approximable by 2-NN, and those for which such guarantees can not be made. For the domain $\Omega = \mathbb{S}^{n-1}$, the difference between them depends on the behavior of eigenvalues of degree $k$ of the integral operator $O_K$. Let $\mu_i$ be an eigenvalue of rank $i$ in a set of eigenvalues of $O_K$ listed in decreasing order (counting multiplicities).

An empirical method that distinguishes between these two classes of activation functions is based on drawing a scatter plot and making a linear regression between $\log(i)$ and $\log(\hat{\mu}_i)$, where $\hat{\mu}_i$ is an eigenvalue of rank $i$ of the empirical kernel matrix $[K^{\text{emp}}(\mathbf{x}_i, \mathbf{x}_j)]_{i,j=1}^N$ where $K^{\text{emp}}(\mathbf{x}, \mathbf{y}) =$

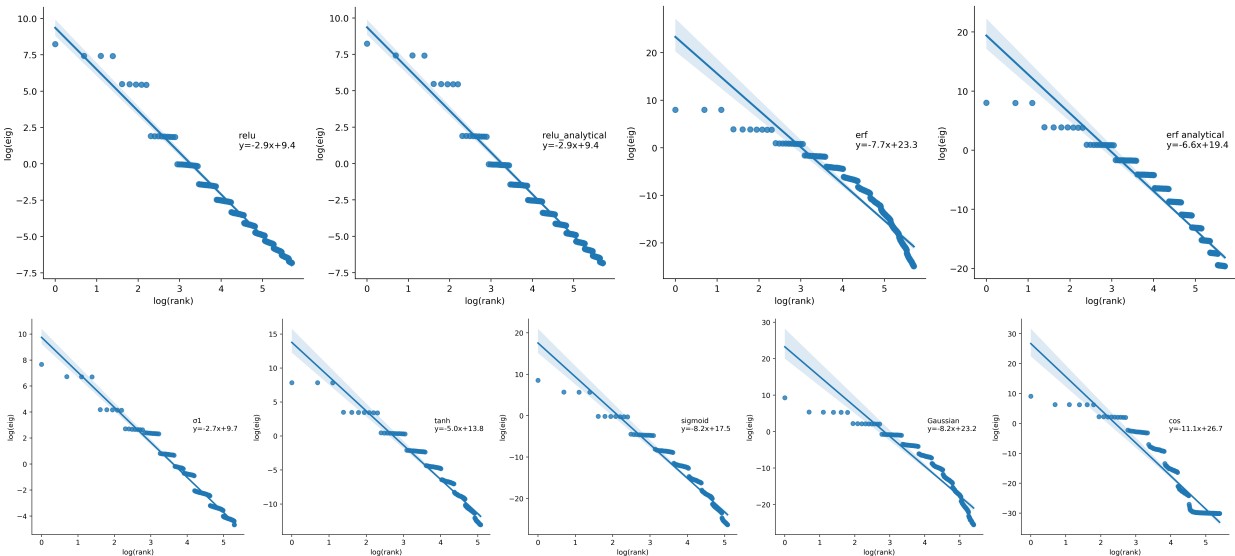

Figure 2: $\log(\hat{\mu}_i)$ versus $\log(i)$ scatter plots for different activation functions with linear regression lines. For relu and erf, eigenvalues of analytically computed NNGP kernels are given for comparison.

$\frac{1}{M}\sum_{i=1}^{M}\sigma(\boldsymbol{\omega}_i^\top \mathbf{x})\sigma(\boldsymbol{\omega}_i^\top \mathbf{y})$ and $\{\boldsymbol{\omega}_i\}_1^M$ are sampled according to $\mathcal{N}(\mathbf{0}, I_n)$, $\{\mathbf{x}_i\}_1^N$ are sampled uniformly on a sphere $\mathbb{S}^{n-1}$. A justification of this method is based on the fact that $\hat{\mu}_i \approx \mu_i$ if $i$ is substantially smaller than $N$ Braun [2006] and $M$ is chosen large enough to estimate the NNGP kernel accurately. In our experiments we set $M = N = 20000$. Since the multiplicity of an eigenvalue of order $k$ is $N(n,k)$, a list of $\mu_i$'s should contain segments of equal eigenvalues. We observe this pattern in empirical $\hat{\mu}_i$'s at the beginning of their list. This allowed us (without any substantiation) to use the following rule of thumb to identify the number of eigenvalues to be included in a training set for linear regression: as eigenvalues which we associate with some order $k$ align into a group with an angle of inclination smaller than $\frac{\pi}{4}$, we assume them to be close to theoretical values.

For popular activation functions and $n = 3$, our results are given in Figure 2. For comparison, we also give 2 plots (for ReLU and erf) for which eigenvalues were computed from an empirical kernel matrix but the kernel function itself was given by an analytical formula. As expected, scatter plots for ReLU and $\sigma_1(x) = \text{ReLU}(x) - \text{ReLU}(x-1)$ are almost identical. Since eigenvalues satisfy $\lambda_k \asymp^n k^{-n}$ for ReLU Geifman et al. [2020], it is natural to conjecture that the same decay rate holds for $\sigma_1$ too. Exponential decay rates for the Gaussian and the cosine activation functions are proved in Appendix H and Appendix I, and scatter plots fully verify those estimates. For the sigmoid and the hyperbolic tangent functions, eigenvalues seem also to decay exponentially, though our interpretation of scatter plots is indecisive due to the lack of any other evidence on the form of the NNGP in that case.

To summarize, we include ReLU and $\sigma_1$ in the first class and the Gaussian, the cosine, and the sine (and likely, sigmoid and tanh) in the second class. Note that the multiplicity of an eigenvalue of order $k$, i.e. of $\lambda_k$, is $N(n,k) \asymp^n k^{n-2}$. Therefore, if $\lambda_k \asymp^n k^{-n-\frac{2}{3}}$, then an eigenvalue of rank $i$ asymptotically behaves like $\mu_i \asymp^n i^{-\frac{n+\frac{2}{3}}{n-1}}$. Activation functions of the first class should have an absolute value of the slope of the regression function smaller than $\frac{n+\frac{2}{3}}{n-1}$, or $\frac{11}{6} \approx 1.83$ for $n = 3$. Definitely, a plot for ReLU should have the slope $-\frac{n}{n-1} = -1.5$, due to the fact that $\lambda_k \asymp^n k^{-n}$. This is in tension with the first two scatter plots of Figure 2 where the slope is larger, i.e. 2.7-2.9. We attribute this to the insufficiency in the number of accurately computed eigenvalues, i.e. probably the slope decreases slightly for larger ranks.

**Learnability of $Y_k$ by random features model (RFM).** In the proof of Theorem 5 a lower bound on $C_{Y_k,\mathbb{S}^{n-1}}$ was given for a certain function $Y_k$. The function itself was described in the proof of Lemma 9. It can be simply defined by $Y_k = \sum_{j=1}^{N(n,k)} x_j Y_{k,j}$ where $\mathbf{x} \in \mathbb{R}^{N(n,k)}$ is a random vector distributed according to the uniform distribution on $\mathbb{S}^{n-1}$.

We experimented with the learnability of $Y_k$ by random features model (RFM), i.e. a 2-NN with a single layer of hidden neurons in which only the output layer's weights are trained (in fact, they are also not trained, but analytically computed using a linear regression formula). Also, we experimented with an optimized RFM (RFM+opt), which is a method in which we first compute weights by RFM and afterward train weights (of both the first and the second layer) by Adam. According to the inequality (4), the square of the $\mathcal{H}_{\Sigma^{(1)}}$-norm is proportional to the number of neurons that is enough to

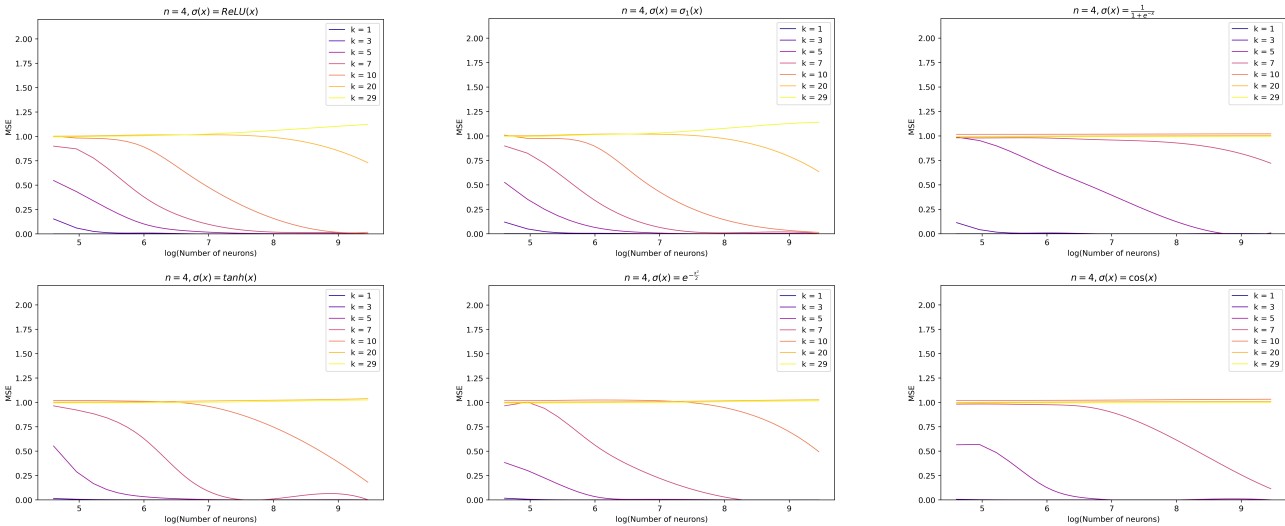

Figure 3: Achieved MSE when learning $Y_k$ by random features model as a function of the number of hidden neurons ($n = 4$). Pictures for other $n$ can be found in the Appendix.

approximate the target function (also, from the construction of Theorem 4 it is clear that $\frac{\|Y_k\|^2_{\mathcal{H}_{\Sigma^{(1)}}}}{\varepsilon^2}$ is an upper bound on the number of neurons needed for RFM to $\varepsilon$-approximate the target function). Since eigenvalues of ReLU ($\sigma_1$, Tanh, sigmoid) NNGP kernel $\Sigma^{(1)}$ decay slower than in the case of the cosine/gaussian activation, $Y_k$ has a smaller RKHS norm ($\|Y_k\|_{\mathcal{H}_{\Sigma^{(1)}}} = \frac{1}{\lambda_k}$), and it is natural to expect that $Y_k$ will be better approximated by the first type of networks than by cosine/gaussian networks. As Figure 3 shows, this is indeed the case for RFM (figures for RFM+opt can be found in the Appendix and they show that the role of the initialization step fade away as we train all weights). This means that our separation of activation functions into two classes can be understood in the following way. For the first class of activations ($\lambda_k \gg k^{-n-\frac{2}{3}} \log^{1/2} k$), RFM often allows to simply construct an approximation that is better than the one that is guaranteed by Barron's theorem. For the second class of activations, this cannot be done by RFM. Note that, unlike $\mathcal{H}_{\Sigma^{(1)}}$-norm, Barron's norm does not depend on $\sigma$. Unlike RFM, Barron's approximation is quite non-constructive. For the second type of activation function, it is an interesting open problem how to simply and "without any optimization" approximate a function better than Barron's approximation.

**Learnability of $Y_k$ by gradient-based methods: a tension with the NTK theory.** For the domain $\mathbf{\Omega} = \mathbb{S}^{n-1}$ not only the NNGP kernel is zonal, but also the NTK is. Therefore, theorems 5 and 6 can be applied to the NTK. Let us denote the NTK of a 2-layer NN by $K$. According to Remark 4, both the norm of $Y_k$ in the Barron space and the norm of $Y_k$ in $\mathcal{H}_K$ grow very rapidly with $k$. In other words, neither Barron's theorem nor Theorem 4 guarantees the existence of a short 2-NN that approximates $Y_k$. Moreover, according

to the NTK theory, this function must be a hard target for the gradient descent training of 2-NNs, in the infinite width limit. Let us show that.

Recall that 2-NNs trained by the gradient descent, in the infinite width limit, with a weight vector properly initialized to $\mathbf{w}_0$ and with a regularization term $\lambda \|\mathbf{w} - \mathbf{w}_0\|^2$, are equivalent to the Kernel Ridge Regression, i.e. to the optimization task $\min_{f \in \mathcal{H}_K} \mathrm{MSE}(f) + \lambda \|f\|^2_{\mathcal{H}_K}$ Hu et al. [2020]. Therefore, a large norm of $Y_k$ in $\mathcal{H}_K$ means that $\lambda$ must be a small parameter for such a 2-NN to succeed, or alternatively, the optimal weight vector should be located far from the initialization $\mathbf{w}_0$.

Those considerations make us expect that this function is hard to approximate by a 2-NN, and is especially hard to learn if an activation function has the NTK eigenvalues decreasing exponentially fast (like the Gaussian or the cosine functions). However, our experiments show that $Y_k$ can be successfully trained by gradient-based methods. Moreover, the performance of different activation functions contradicts the NTK theory. Certainly, this outcome is due to the finiteness of real neural networks.

A synthetic supervised dataset with $Y_k$ as a target function was generated. A loss function to minimize was set to the mean squared error (MSE), and an optimization algorithm was set to the Adam optimizer with the learning rate 0.01 Kingma and Ba [2015]. We experimented with the number of neurons of a hidden layer equal to 256 and 1024. On Figure 4 one can see the behavior of the loss function (averaged over 5 independent repeated experiments) during the training process for 2-NNs with activation functions (a) $\sigma(x) = e^{-\frac{x^2}{2}}$, (b) $\sigma(x) = \cos(x)$, (c) $\sigma(x) = \mathrm{ReLU}(x)$.

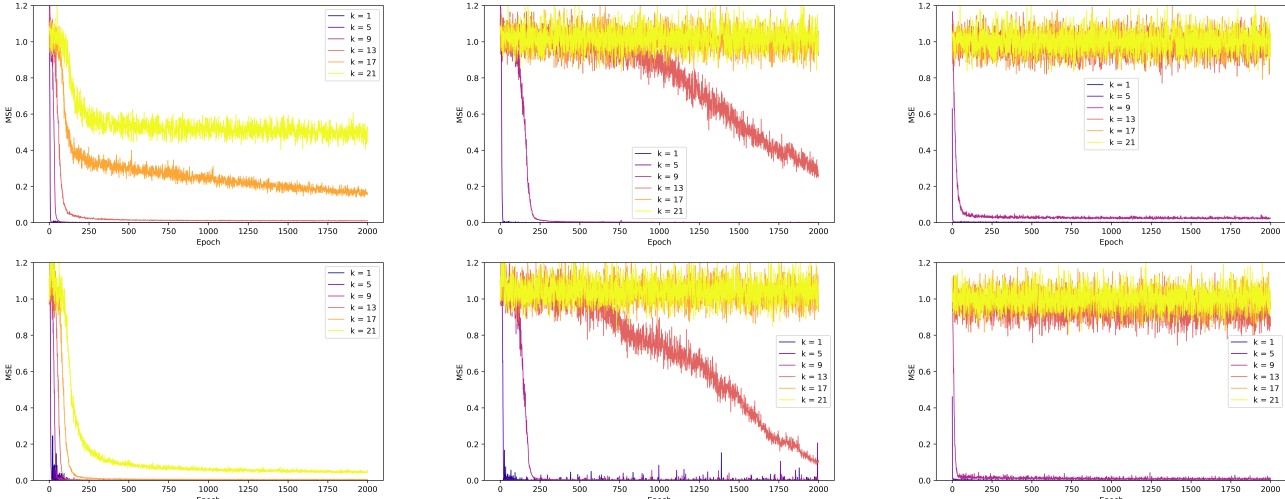

Figure 4: MSE dynamics during learning $Y_k$ by a 2-NN with the number of hidden neurons 256 and 1024 (rows) and an activation function (columns): (a) $\sigma(x) = e^{-\frac{x^2}{2}}$, (b) $\sigma(x) = \cos(x)$, (c) $\sigma(x) = \mathrm{ReLU}(x)$.

As we see, the Gaussian function outperforms the cosine function, and the cosine function outperforms the ReLU for all orders $k$. The achieved MSE for the Gaussian function is non-trivial (i.e. smaller than the baseline 1.0 corresponding to a trained zero function) for all $k = 1, \cdots, 21$, while the cosine function fails for $k \geq 14$ and the ReLU fails for $k \geq 10$. Plots are given for $n = 3$, though we report very similar results for spherical harmonics in higher dimensions (using a code with precomputed spherical harmonics Dutordoir et al. [2020]). Possibly, looking into the case of $n = 2$ allows us to explain the described picture. In that case, $\mathbb{S}^1$ is isomorphic to $[0, 2\pi]$ with its endpoints identified, $L_2(\mathbb{S}^1)$ corresponds to periodic functions on $[0, 2\pi]$, and $Y_k$ can be given as $Y_k(x) = \cos(kx + \phi)$. Therefore, it is not surprising that $Y_k$ can be trained by a 2-NN with the cosine activation function or the Gaussian function (the latter is capable of approximating cosine's waves).

To summarize, we conclude that even if a function's norms blow up in $\mathcal{H}_K$, this does not necessarily imply its hardness as a target for the gradient descent training of 2-NNs. These figures demonstrate that the approximation theorems that we analyzed are responsible only for certain aspects of the approximation power of NNs. Moreover, the Neural Tangent Kernel theory definitely does not explain the learnability of functions by 2-NNs with a finite width of hidden layers.

Other experiments can be found in the Appendix. Our code is available on github to facilitate the reproducibility of the results.

# 6 CONCLUSIONS

The paper is dedicated to an approximation theory of multi-layer feedforward neural networks based on the NNGP ker-

nel. We show that if a function has a moderate norm in the RKHS defined by the NNGP kernel, then it can be successfully approximated by a corresponding NN.

Besides this, we compare two functional norms, the Barron norm and the RKHS norm of zonal kernels. We classified all activation functions into two groups, those for which the norm in the Barron space is not dominated by the RKHS norm, and those for which the opposite is true. We gave examples of activation functions for both classes. We observed that random spherical harmonics of order $k$ have large norms in both spaces, yet are very well learnable by gradient-based methods with realistic neural networks. It is a topic of future research to study theoretically why such functions are accurately approximable and easily learnable by practical NNs.

# ACKNOWLEDGMENTS

This research has been funded by Nazarbayev University under Faculty-development competitive research grants program for 2023-2025 Grant #20122022FD4131, PI R. Takhanov.

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

# Multi-layer random features and the approximation power of neural networks (Supplementary Material)

**Rustem Takhanov**[1]

[1]Mathematics Dept., Nazarbayev University, Astana, Kazakhstan

## A  PROOF OF THEOREM 1

*Proof.* By construction,

$$\tilde{\Sigma}^{(L)}(\mathbf{x}, \mathbf{x}') = \mathbb{E}_{W^{(1)}, \cdots, W^{(L)}}[\alpha^{(L)}(\mathbf{x}, \theta)\alpha^{(L)}(\mathbf{x}', \theta)]. \tag{6}$$

where $W_{ij}^{(1)} \sim \mathcal{N}(0, 1)$ and $W_{ij}^{(h)} \sim \mathcal{N}(0, \frac{1}{n_{h-1}}), h = 2, \cdots, L$.

Let $L_2(\theta)$ denote the Hilbert space of real-valued functions on $\prod_{h=1}^{L-1} \mathbb{R}^{n_h \times n_{h-1}} \times \mathbb{R}^{n_L}$ with the inner product $\langle f, g \rangle_{L_2(\theta)} = \mathbb{E}_{W^{(1)}, \cdots, W^{(L)}}[f(W^{(1)}, \cdots, W^{(L)})g(W^{(1)}, \cdots, W^{(L)})]$. The latter object is simply a weighted $L_2$-space.

Let $\mathcal{H}_0$ be a span of $\{\tilde{\Sigma}^{(L)}(\mathbf{x}, \cdot)\}_{\mathbf{x} \in \mathbf{\Omega}}$ equipped with the inner product $\langle \sum_{i=1}^k a_i \tilde{\Sigma}^{(L)}(\mathbf{x}_i, \cdot), \sum_{i=1}^l b_i \tilde{\Sigma}^{(L)}(\mathbf{y}_i, \cdot) \rangle_{\mathcal{H}_0} = \sum_{i=1}^k \sum_{j=1}^l a_i b_j \tilde{\Sigma}^{(L)}(\mathbf{x}_i, \mathbf{y}_j)$. Now let us assume that $f \in \mathcal{H}_{\tilde{\Sigma}^{(L)}}$. Recall that $\mathcal{H}_{\tilde{\Sigma}^{(L)}}$ is a completion $\mathcal{H}_0$, therefore, we have

$$\forall \mathbf{x} \in \mathbf{\Omega}, f(\mathbf{x}) = \lim_{i \to +\infty} f_i(\mathbf{x}),$$

where $f_i = \sum_{j=1}^{m_i} a_{ij} \tilde{\Sigma}^{(L)}(\mathbf{x}_{ij}, \cdot)$ and $\lim_{i \to +\infty} \sup_{p \in \mathbb{N}} \|f_{i+p} - f_i\|_{\mathcal{H}_0} = 0$. Using (6) we conclude that the Cauchy sequence $f_i = \sum_{j=1}^{m_i} a_{ij} \tilde{\Sigma}^{(L)}(\mathbf{x}_{ij}, \cdot)$ satisfies

$$\langle f_i, f_{i'} \rangle_{\mathcal{H}_{\tilde{\Sigma}^{(L)}}} = \sum_{j=1}^{m_i} \sum_{j'=1}^{m_{i'}} a_{ij} a_{i'j'} \mathbb{E}_\theta[\alpha^{(L)}(\mathbf{x}_{ij}, \theta)\alpha^{(L)}(\mathbf{x}_{i'j'}, \theta)] = \langle \sum_{j=1}^{m_i} a_{ij} \alpha^{(L)}(\mathbf{x}_{ij}, \cdot), \sum_{j'=1}^{m_{i'}} a_{i'j'} \alpha^{(L)}(\mathbf{x}_{i'j'}, \cdot) \rangle_{L_2(\theta)}.$$

Thus, $\langle f_i, f_{i'} \rangle_{\mathcal{H}_K} = \langle g_i, g_{i'} \rangle_{L_2(\theta)}$ where

$$g_i = \sum_{j=1}^{m_i} a_{ij} \alpha^{(L)}(\mathbf{x}_{ij}, \cdot).$$

From $\|f_i - f_{i'}\|_{\mathcal{H}_{\tilde{\Sigma}^{(L)}}} = \|g_i - g_{i'}\|_{L_2(\theta)}$ we conclude that $\{g_i\}$ is also a Cauchy sequence, but in $L_2(\theta)$. Let us denote its limit in $L_2(\theta)$ by $g$. Note that $\|g\|_{L_2(\theta)} = \|f\|_{\mathcal{H}_{\tilde{\Sigma}^{(L)}}}$.

Since $\sigma$ is bounded we conclude that $\alpha^{(L)}(\mathbf{x}, \cdot) \in L_2(\theta)$ for any $\mathbf{x} \in \mathbf{\Omega}$. Thus, we conclude

$$\langle g, \alpha^{(L)}(\mathbf{x}, \cdot) \rangle_{L_2(\theta)} = \lim_{k \to +\infty} \langle g_k, \alpha^{(L)}(\mathbf{x}, \cdot) \rangle_{L_2(\theta)} = \lim_{i \to +\infty} \sum_{j=1}^{m_i} a_{ij} \tilde{\Sigma}^{(L)}(\mathbf{x}_{ij}, \mathbf{x}) = f(\mathbf{x}).$$

Thus, we obtained a key integral representation for $f$:

$$f(\mathbf{x}) = \mathbb{E}_{W^{(1)}, \cdots, W^{(L)}}[g(W^{(1)}, \cdots, W^{(L)})\alpha^{(L)}(\mathbf{x}, W^{(1)}, \cdots, W^{(L)})].$$

Let us introduce $T$ independent copies of $\theta$: $\theta_1, \cdots, \theta_T$. We define

$$\tilde{f}(\mathbf{x}, \{\theta_i\}_{i=1}^T) = \frac{1}{T}\sum_{i=1}^T g(\theta_i)\alpha^{(L)}(\mathbf{x}, \theta_i).$$

By construction, $f(\mathbf{x}) = \mathbb{E}_{\theta_i}[\tilde{f}(\mathbf{x}, \{\theta_i\}_{i=1}^T)]$. Further, we bound the variance of the distance between $f$ and $\tilde{f}$ by

$$\mathbb{E}_{\theta_i}\big[\|f - \tilde{f}(\cdot, \{\theta_i\}_{i=1}^T)\|_{L_2(\mathbf{\Omega},\mu)}^2\big] =$$
$$\mathbb{E}_{\theta_i}\big(\langle f, f\rangle_{L_2(\mathbf{\Omega},\mu)} - 2\langle f, \tilde{f}(\cdot, \{\theta_i\}_{i=1}^T)\rangle_{L_2(\mathbf{\Omega},\mu)} + \|\tilde{f}(\cdot, \{\theta_i\}_{i=1}^T)\|_{L_2(\mathbf{\Omega},\mu)}^2\big) =$$
$$\mathbb{E}_{X\sim\mu}\mathbb{E}_{\theta_i}\big[|\tilde{f}(X, \{\theta_i\}_{i=1}^T)|^2\big] - \langle f, f\rangle_{L_2(\mathbf{\Omega},\mu)} = \mathbb{E}_{X\sim\mu}\big[\mathrm{Var}_{\theta_i}[\tilde{f}(X, \{\theta_i\}_{i=1}^T) \mid X]\big].$$

One can unfold $\mathrm{Var}_{\theta_i}[\tilde{f}(X, \{\theta_i\}_{i=1}^T) \mid X]$ in the following way:

$$\mathrm{Var}_{\theta_i}\big[\frac{1}{T}\sum_{i=1}^T g(\theta_i)\alpha^{(L)}(X, \theta_i) \mid X\big] = \frac{\mathrm{Var}_{\theta_i}[g(\theta_i)\alpha^{(L)}(X, \theta_i) \mid X]}{T}.$$

Using boundedness of $\sigma$ and $\mathbb{E}_{\theta_i}\big[|g(\theta_i)|^2\big] = \|f\|_{\mathcal{H}_{\tilde{\Sigma}(L)}}^2$ we conclude that

$$\mathrm{Var}_\theta[g(\theta_i)\alpha^{(L)}(X, \theta_i)] \leq \mathbb{E}[|g(\theta_i)\alpha^{(L)}(X, \theta_i)|^2] \leq \|\sigma\|_\infty^2\|f\|_{\mathcal{H}_{\tilde{\Sigma}(L)}}^2,$$

and

$$\mathbb{E}_{\theta_i}\big[\|f - \tilde{f}(\cdot, \{\theta_i\}_{i=1}^{n_{L+1}})\|_{L_2(\mathbf{\Omega},\mu)}^2\big] = \mathbb{E}_{X\sim\mu}\big[\mathrm{Var}_{\theta_i}[\tilde{f}(X, \{\theta_i\}_{i=1}^{n_{L+1}}) \mid X]\big] \leq \frac{\|\sigma\|_\infty^2\|f\|_{\mathcal{H}_{\tilde{\Sigma}(L)}}^2}{T}.$$

Since the latter expected value is smaller than $\frac{\|\sigma\|_\infty^2\|f\|_{\mathcal{H}_{\tilde{\Sigma}(L)}}^2}{T}$, then there exist $\{\theta_i\}$ such that $\|f - \tilde{f}(\cdot, \{\theta_i\}_{i=1}^T)\|_{L_2(\mathbf{\Omega},\mu)}^2 \leq \frac{\|\sigma\|_\infty^2\|f\|_{\mathcal{H}_{\tilde{\Sigma}(L)}}^2}{T}$, from which the statement of the theorem follows directly. $\qquad\square$

# B  PROOF OF THEOREM 2: CONCENTRATION OF $\Sigma_{\mathrm{emp}}^{(h)}(\mathbf{x}, \mathbf{x}')$ AROUND ITS MEAN

Let us define $\mathcal{M}_+ \subseteq \mathbb{R}^{2\times 2}$ as a set of positive definite $2 \times 2$-matrices. For a given function $\psi : \mathbb{R} \to \mathbb{R}$, let us introduce the mapping $\overline{\psi} : \mathcal{M}_+ \to \mathbb{R}$ by $\overline{\psi}(\Sigma) = \mathbb{E}_{(u,v)\sim\mathcal{N}(\mathbf{0},\Sigma)}[\psi(u)\psi(v)]$. For completeness, properties of $\overline{\psi}$ that we will need (with their proofs) are given in Section E.

Let us denote

$$\gamma^{(h)} = \sup_{\mathbf{x},\mathbf{x}'} \mathrm{Var}[\Sigma_{\mathrm{emp}}^{(h)}(\mathbf{x}, \mathbf{x})] + \mathrm{Var}[\Sigma_{\mathrm{emp}}^{(h)}(\mathbf{x}', \mathbf{x}')] + 2\mathrm{Var}[\Sigma_{\mathrm{emp}}^{(h)}(\mathbf{x}, \mathbf{x}')]. \tag{7}$$

**Lemma 1.** *For $h = 0, \cdots, L-1$, we have*

$$\mathrm{Var}[\Sigma_{\mathrm{emp}}^{(h+1)}(\mathbf{x}, \mathbf{x}')] \leq \frac{\mathbb{E}[\overline{\sigma^2}(\Lambda_{\mathrm{emp}}^{(h)}(\mathbf{x}, \mathbf{x}'))]}{n_{h+1}} + \mathrm{Var}[\overline{\sigma}(\Lambda_{\mathrm{emp}}^{(h)}(\mathbf{x}, \mathbf{x}'))].$$

*Proof.* $\Sigma_{\mathrm{emp}}^{(h+1)}(\mathbf{x}, \mathbf{x}')$ can be represented as

$$\Sigma_{\mathrm{emp}}^{(h+1)}(\mathbf{x}, \mathbf{x}') = \frac{1}{n_{h+1}}\sum_{i=1}^{n_{h+1}} \sigma(\sum_{j=1}^{n_h} W_{ij}^{(h+1)}\alpha_j^{(h)}(\mathbf{x}, \theta))\sigma(\sum_{j=1}^{n_h} W_{ij}^{(h+1)}\alpha_j^{(h)}(\mathbf{x}', \theta)).$$

Given $W_1, \cdots, W_h$, $\sigma(\sum_{j=1}^{n_h} W_{ij}^{(h+1)}\alpha_j^{(h)}(\mathbf{x}, \theta))$ are independent for different $i = 1, \cdots, n_{h+1}$. Therefore,

$$\mathrm{Var}[\Sigma_{\mathrm{emp}}^{(h+1)}(\mathbf{x}, \mathbf{x}') \mid W_1, \cdots, W_h] =$$
$$\frac{1}{n_{h+1}^2}\sum_{i=1}^{n_{h+1}} \mathrm{Var}[\sigma(\sum_{j=1}^{n_h} W_{ij}^{(h+1)}\alpha_j^{(h)}(\mathbf{x}, \theta))\sigma(\sum_{j=1}^{n_h} W_{ij}^{(h+1)}\alpha_j^{(h)}(\mathbf{x}', \theta)) \mid W_1, \cdots, W_h] =$$
$$\frac{1}{n_{h+1}}(\overline{\sigma^2}(\Lambda_{\mathrm{emp}}^{(h)}(\mathbf{x}, \mathbf{x}')) - \overline{\sigma}(\Lambda_{\mathrm{emp}}^{(h)}(\mathbf{x}, \mathbf{x}'))^2) \leq \frac{\overline{\sigma^2}(\Lambda_{\mathrm{emp}}^{(h)}(\mathbf{x}, \mathbf{x}'))}{n_{h+1}}.$$

By the law of total variance, we have

$$\text{Var}[\Sigma_{\text{emp}}^{(h+1)}(\mathbf{x},\mathbf{x}')] = \mathbb{E}_{W^{(1)},\cdots,W^{(h)}}[\text{Var}[\Sigma_{\text{emp}}^{(h+1)}(\mathbf{x},\mathbf{x}') \mid W^{(1)},\cdots,W^{(h)}]]+$$
$$\text{Var}_{W^{(1)},\cdots,W^{(h)}}[\mathbb{E}[\Sigma_{\text{emp}}^{(h+1)}(\mathbf{x},\mathbf{x}') \mid W^{(1)},\cdots,W^{(h)}]].$$

From the former, we conclude that the first term is bounded by $\frac{\mathbb{E}[\overline{\sigma^2}(\Lambda_{\text{emp}}^{(h)}(\mathbf{x},\mathbf{x}'))]}{n_{h+1}}$. The expression inside the second term, by construction, is

$$\mathbb{E}[\Sigma_{\text{emp}}^{(h+1)}(\mathbf{x},\mathbf{x}') \mid W^{(1)},\cdots,W^{(h)}] = \overline{\sigma}(\Lambda_{\text{emp}}^{(h)}(\mathbf{x},\mathbf{x}')).$$

After we plug in $\overline{\sigma}(\Lambda_{\text{emp}}^{(h)}(\mathbf{x},\mathbf{x}'))$ into the second term we obtain the needed inequality. $\qquad\square$

By construction, $|\overline{\sigma^2}(\Sigma)| \leq \|\sigma\|_\infty^4$, therefore, the first term in the latter lemma is bounded by $\frac{\|\sigma\|_\infty^4}{n_{h+1}}$. From Lemma 8 we obtain that $\overline{\sigma}(\Sigma)$ is Lipschitz w.r.t. to the Frobenius norm if $\sigma$, $\sigma'$ and $\sigma''$ are all bounded. The following lemma specifies our bound for such activation functions $\sigma$.

**Lemma 2.** *If $\overline{\sigma^2}$ is bounded by $c_1$ and $\overline{\sigma}$ is $c_2$-Lipschitz w.r.t. the Frobenius norm, then*

$$\text{Var}[\Sigma_{\text{emp}}^{(h+1)}(\mathbf{x},\mathbf{x}')] \leq \frac{c_1}{n_{h+1}} + c_2^2\gamma^{(h)}.$$

*Proof.* Let us denote by $\Lambda_{\text{emp}-c}^{(h)}(\mathbf{x},\mathbf{x}')$ an independent copy of $\Lambda_{\text{emp}}^{(h)}(\mathbf{x},\mathbf{x}')$. Then, from $c_2$-Lipschitzness of $\overline{\sigma}$ we obtain

$$\text{Var}[\overline{\sigma}(\Lambda_{\text{emp}}^{(h)}(\mathbf{x},\mathbf{x}'))] = \frac{1}{2}\mathbb{E}[\left(\overline{\sigma}(\Lambda_{\text{emp}}^{(h)}(\mathbf{x},\mathbf{x}')) - \overline{\sigma}(\Lambda_{\text{emp}-c}^{(h)}(\mathbf{x},\mathbf{x}'))\right)^2] \leq$$

$$\frac{c_2^2}{2}\mathbb{E}[\|\Lambda_{\text{emp}}^{(h)}(\mathbf{x},\mathbf{x}') - \Lambda_{\text{emp}-c}^{(h)}(\mathbf{x},\mathbf{x}')\|_F^2] = c_2^2\text{Var}[\Sigma_{\text{emp}}^{(h)}(\mathbf{x},\mathbf{x})] + c_2^2\text{Var}[\Sigma_{\text{emp}}^{(h)}(\mathbf{x}',\mathbf{x}')] + 2c_2^2\text{Var}[\Sigma_{\text{emp}}^{(h)}(\mathbf{x},\mathbf{x}')].$$

After we plug in the latter bound into the R.H.S. of the previous lemma, we obtain the needed statement. $\qquad\square$

*Proof of Theorem 2..* The previous lemma, together with Lemma 8, indicates that $\gamma^{(h+1)}$ satisfies

$$\gamma^{(h+1)} \leq \frac{4c_1}{n_{h+1}} + 4c_2^2\gamma^{(h)}.$$

where $c_1 = \|\sigma\|_\infty^4$ and $c_2 = \max(\|\sigma''\|_\infty\|\sigma\|_\infty, \|\sigma'\|_\infty^2)$.

Since $\gamma^{(0)} = 0$, by applying the latter $h$ times we obtain

$$\gamma^{(h)} \leq 4c_1\left(\frac{1}{n_h} + \frac{4c_2^2}{n_{h-1}} + \cdots + \frac{(4c_2^2)^{h-1}}{n_1}\right). \tag{8}$$

Finally,

$$2\text{Var}[\Sigma_{\text{emp}}^{(h)}(\mathbf{x},\mathbf{x}')] \leq \gamma^{(h)} \leq 4c_1\left(\frac{1}{n_h} + \frac{4c_2^2}{n_{h-1}} + \cdots + \frac{(4c_2^2)^{h-1}}{n_1}\right),$$

and the proof is completed. $\qquad\square$

# C   PROOF OF THEOREM 3: AN APPROXIMATION OF $\tilde{\Sigma}^{(h)}(\mathbf{x},\mathbf{x}')$ BY $\Sigma^{(h)}(\mathbf{x},\mathbf{x}')$

Note that $\Sigma^{(h+1)}(\mathbf{x},\mathbf{x}') = \overline{\sigma}(\Lambda^{(h)}(\mathbf{x},\mathbf{x}'))$. Relationship between their finite versions, i.e. $\tilde{\Sigma}^{(h+1)}(\mathbf{x},\mathbf{x}')$ and $\tilde{\Sigma}^{(h)}(\mathbf{x},\mathbf{x}')$ is trickier.

**Lemma 3.** *If $\overline{\sigma}$ is twice continuously differentiable and $|\frac{\partial^2\overline{\sigma}(\Sigma)}{\partial\Sigma_{a,b}\partial\Sigma_{c,d}}| \leq C$, we have*

$$|\tilde{\Sigma}^{(h+1)}(\mathbf{x},\mathbf{x}') - \overline{\sigma}(\tilde{\Lambda}^{(h)}(\mathbf{x},\mathbf{x}'))| \leq 4C\gamma^{(h)}.$$

*Proof.* We have

$$\tilde{\Sigma}^{(h+1)}(\mathbf{x}, \mathbf{x}') = \mathbb{E}[\sigma(\sum_{j=1}^{n_h} W_{ij}^{(h+1)}\alpha_j^{(h)}(\mathbf{x}, \theta))\sigma(\sum_{j=1}^{n_h} W_{ij}^{(h+1)}\alpha_j^{(h)}(\mathbf{x}', \theta))] =$$

$$\mathbb{E}_{W^{(1)}, \cdots, W^{(h)}}\big[\mathbb{E}_{W^{(h+1)}}[\sigma(\sum_{j=1}^{n_h} W_{ij}^{(h+1)}\alpha_j^{(h)}(\mathbf{x}, \theta))\sigma(\sum_{j=1}^{n_h} W_{ij}^{(h+1)}\alpha_j^{(h)}(\mathbf{x}', \theta)) \mid W^{(1)}, \cdots, W^{(h)}]\big] =$$

$$\mathbb{E}_{W^{(1)}, \cdots, W^{(h)}}[\overline{\sigma}(\Lambda_{\text{emp}}^{(h)}(\mathbf{x}, \mathbf{x}'))].$$

Since $\overline{\sigma}$ is twice continuously differentiable, we have

$$\overline{\sigma}(\Lambda_{\text{emp}}^{(h)}(\mathbf{x}, \mathbf{x}')) = \overline{\sigma}(\tilde{\Lambda}^{(h)}(\mathbf{x}, \mathbf{x}')) + \langle \frac{\partial \overline{\sigma}}{\partial \Sigma}(\tilde{\Lambda}^{(h)}(\mathbf{x}, \mathbf{x}')), \Lambda_{\text{emp}}^{(h)}(\mathbf{x}, \mathbf{x}') - \tilde{\Lambda}^{(h)}(\mathbf{x}, \mathbf{x}')\rangle +$$

$$\sum_{\{a,b\}\in\{\mathbf{x},\mathbf{x}'\}} \sum_{\{c,d\}\in\{\mathbf{x},\mathbf{x}'\}} C_{(a,b),(c,d)}(\Sigma_{\text{emp}}^{(h)}(a, b) - \tilde{\Sigma}^{(h)}(a, b))(\Sigma_{\text{emp}}^{(h)}(c, d) - \tilde{\Sigma}^{(h)}(c, d)),$$

where $C_{(a,b),(c,d)} = \int_0^1 \frac{\partial^2 \overline{\sigma}(t\Sigma_{\text{emp}}^{(h)} + (1-t)\tilde{\Sigma}^{(h)})}{\partial \Sigma_{a,b} \partial \Sigma_{c,d}}(1-t)dt$. Since $|\frac{\partial^2 \overline{\sigma}(t\Sigma_{\text{emp}}^{(h)} + (1-t)\tilde{\Sigma}^{(h)})}{\partial \Sigma_{a,b} \partial \Sigma_{c,d}}| \le C$ we have $|C_{(a,b),(c,d)}| \le C$. Also, a spectral norm of any symmetric matrix $[a_{ij}] \in \mathbb{R}^{4\times 4}$ does not exceed $4\max a_{ij}$. Thus, we conclude that

$$-4C\|\Lambda_{\text{emp}}^{(h)}(\mathbf{x}, \mathbf{x}') - \tilde{\Lambda}^{(h)}(\mathbf{x}, \mathbf{x}')\|_F^2 \le$$

$$\overline{\sigma}(\Lambda_{\text{emp}}^{(h)}(\mathbf{x}, \mathbf{x}')) - \overline{\sigma}(\tilde{\Lambda}^{(h)}(\mathbf{x}, \mathbf{x}')) - \langle \frac{\partial \overline{\sigma}}{\partial \Sigma}(\tilde{\Lambda}^{(h)}(\mathbf{x}, \mathbf{x}')), \Lambda_{\text{emp}}^{(h)}(\mathbf{x}, \mathbf{x}') - \tilde{\Lambda}^{(h)}(\mathbf{x}, \mathbf{x}')\rangle \le$$

$$4C\|\Lambda_{\text{emp}}^{(h)}(\mathbf{x}, \mathbf{x}') - \tilde{\Lambda}^{(h)}(\mathbf{x}, \mathbf{x}')\|_F^2.$$

After we apply the expectation to all sides of the inequality we obtain

$$|\mathbb{E}[\overline{\sigma}(\Lambda_{\text{emp}}^{(h)}(\mathbf{x}, \mathbf{x}'))] - \overline{\sigma}(\tilde{\Lambda}^{(h)}(\mathbf{x}, \mathbf{x}'))| \le 4C\mathbb{E}[\|\Lambda_{\text{emp}}^{(h)}(\mathbf{x}, \mathbf{x}') - \tilde{\Lambda}^{(h)}(\mathbf{x}, \mathbf{x}')\|_F^2] = 4C\gamma^{(h)}.$$

Therefore,

$$|\tilde{\Sigma}^{(h+1)}(\mathbf{x}, \mathbf{x}') - \overline{\sigma}(\tilde{\Lambda}^{(h)}(\mathbf{x}, \mathbf{x}'))| \le 4C\gamma^{(h)}.$$

$\square$

Let us denote $\sup_{\mathbf{x},\mathbf{x}'} 2|\tilde{\Sigma}^{(h)}(\mathbf{x}, \mathbf{x}') - \Sigma^{(h)}(\mathbf{x}, \mathbf{x}')| + |\tilde{\Sigma}^{(h)}(\mathbf{x}, \mathbf{x}) - \Sigma^{(h)}(\mathbf{x}, \mathbf{x})| + |\tilde{\Sigma}^{(h)}(\mathbf{x}', \mathbf{x}') - \Sigma^{(h)}(\mathbf{x}', \mathbf{x}')|$ by $\tilde{\gamma}^{(h)}$.

**Lemma 4.** *If $\overline{\sigma}$ is twice continuously differentiable with $|\frac{\partial \overline{\sigma}(\Sigma)}{\partial \Sigma_{a,b}}| \le c$ and $|\frac{\partial^2 \overline{\sigma}(\Sigma)}{\partial \Sigma_{a,b}\partial \Sigma_{c,d}}| \le C$, then*

$$\tilde{\gamma}^{(h)} \le 16C \sum_{i=1}^{h-1}(10c)^{i-1}\gamma^{(h-i)}.$$

*Proof.* From Lemma 3 and $\Sigma^{(h+1)}(\mathbf{x}, \mathbf{x}') = \overline{\sigma}(\Lambda^{(h)}(\mathbf{x}, \mathbf{x}'))$ we obtain

$$|\tilde{\Sigma}^{(h+1)}(\mathbf{x}, \mathbf{x}') - \Sigma^{(h+1)}(\mathbf{x}, \mathbf{x}')| \le 4C\gamma^{(h)} + |\overline{\sigma}(\tilde{\Lambda}^{(h)}(\mathbf{x}, \mathbf{x}')) - \overline{\sigma}(\Lambda^{(h)}(\mathbf{x}, \mathbf{x}'))| \le$$

$$4C\gamma^{(h)} + c(2|\tilde{\Sigma}^{(h)}(\mathbf{x}, \mathbf{x}') - \Sigma^{(h)}(\mathbf{x}, \mathbf{x}')| + |\tilde{\Sigma}^{(h)}(\mathbf{x}, \mathbf{x}) - \Sigma^{(h)}(\mathbf{x}, \mathbf{x})| + |\tilde{\Sigma}^{(h)}(\mathbf{x}', \mathbf{x}') - \Sigma^{(h)}(\mathbf{x}', \mathbf{x}')|) =$$

$$4C\gamma^{(h)} + c\tilde{\gamma}^{(h)}.$$

For $a \in \{\mathbf{x}, \mathbf{x}'\}$ we have

$$|\tilde{\Sigma}^{(h+1)}(a, a) - \Sigma^{(h+1)}(a, a)| \le 4C\gamma^{(h)} + 4c|\tilde{\Sigma}^{(h)}(a, a) - \Sigma^{(h)}(a, a)| \le 4C\gamma^{(h)} + 4c\tilde{\gamma}^{(h)}.$$

Therefore,

$$\tilde{\gamma}^{(h+1)} \le 16C\gamma^{(h)} + 10c\tilde{\gamma}^{(h)},$$

and finally,

$$\tilde{\gamma}^{(h)} \leq 16C(\gamma^{(h-1)} + 10c\gamma^{(h-2)} + (10c)^2\gamma^{(h-3)} + \cdots + (10c)^{h-2}\gamma^{(1)}).$$

$\square$

*Proof of Theorem 3..* Let us set $C = \frac{1}{4}\max(\|\sigma''''\|_\infty\|\sigma\|_\infty, \|\sigma'''\|_\infty\|\sigma'\|_\infty, \|\sigma''\|_\infty^2)$ and $c = \frac{1}{2}\max(\|\sigma\|_\infty\|\sigma''\|_\infty, \|\sigma'\|_\infty^2, \frac{5}{4})$. From Lemma 8 we obtain that $|\frac{\partial\overline{\sigma}(\Sigma)}{\partial\Sigma_{a,b}}| \leq c$ and $|\frac{\partial^2\overline{\sigma}(\Sigma)}{\partial\Sigma_{a,b}\partial\Sigma_{c,d}}| \leq C$. Thus, the previous lemma gives us

$$2|\tilde{\Sigma}^{(h)}(\mathbf{x},\mathbf{x}') - \Sigma^{(h)}(\mathbf{x},\mathbf{x}')| \leq \tilde{\gamma}^{(h)} \leq 16C\sum_{i=1}^{h-1}(10c)^{i-1}\gamma^{(h-i)}.$$

From equation (8) we conclude

$$|\tilde{\Sigma}^{(h)}(\mathbf{x},\mathbf{x}') - \Sigma^{(h)}(\mathbf{x},\mathbf{x}')| \leq 8C\sum_{i=1}^{h-1}(10c)^{i-1}C_1\sum_{j=1}^{h-i}\frac{C_2^{h-i-j}}{n_j} \ll$$

$$\|\sigma\|_\infty^4\max(\|\sigma''''\|_\infty\|\sigma\|_\infty, \|\sigma'''\|_\infty\|\sigma'\|_\infty, \|\sigma''\|_\infty^2)\sum_{j=1}^{h-1}\frac{r_j}{n_j}.$$

where $C_1 = 4\|\sigma\|_\infty^4$, $C_2 = 4\max(\|\sigma''\|_\infty\|\sigma\|_\infty, \|\sigma'\|_\infty^2, \frac{5}{4})^2 = 16c^2$ and $r_j = \sum_{i=1}^{h-j}(10c)^iC_2^{h-j-i}$. We have $\sum_{i=1}^{h-j}(10c)^iC_2^{h-j-i} = (16c^2)^{h-j}\sum_{i=1}^{h-j}(1.6c)^{-i} \leq (16c^2)^{h-j}(h-j)$. Thus,

$$|\tilde{\Sigma}^{(h)}(\mathbf{x},\mathbf{x}') - \Sigma^{(h)}(\mathbf{x},\mathbf{x}')| \ll$$

$$\|\sigma\|_\infty^4\max(\|\sigma''''\|_\infty\|\sigma\|_\infty, \|\sigma'''\|_\infty\|\sigma'\|_\infty, \|\sigma''\|_\infty^2)\sum_{j=1}^{h-1}\frac{\max(2\|\sigma''\|_\infty\|\sigma\|_\infty, 2\|\sigma'\|_\infty^2, \frac{5}{2})^{2h-2j}(h-j)}{n_j}.$$

Setting $h = L$ gives the desired statement. $\square$

# D PROOF OF THEOREM 4

**Lemma 5.** *Let $\mu$ be a probabilistic measure on $\mathbf{\Omega} \subseteq \mathbb{R}^n$. Let $K_1, K_2 : \mathbf{\Omega} \times \mathbf{\Omega} \to \mathbb{R}$ be two Mercer kernels such that $|K_1(\mathbf{x},\mathbf{y}) - K_2(\mathbf{x},\mathbf{y})| < \varepsilon$. Then square roots of operators $\mathrm{O}_{K_1}, \mathrm{O}_{K_2} : L_2(\mathbf{\Omega},\mu) \to L_2(\mathbf{\Omega},\mu)$ where $\mathrm{O}_{K_i}[\phi](\mathbf{x}) = \int_{\mathbf{\Omega}}K_i(\mathbf{x},\mathbf{y})\phi(\mathbf{y})d\mu(\mathbf{y})$ satisfy*

$$\|\mathrm{O}_{K_1}^{1/2} - \mathrm{O}_{K_2}^{1/2}\|_{L_2(\mathbf{\Omega},\mu)\to L_2(\mathbf{\Omega},\mu)} \leq c\varepsilon^{1/2},$$

*where $c$ is some universal constant.*

*Proof.* We have

$$\|\mathrm{O}_{K_1} - \mathrm{O}_{K_2}\|_{L_2(\mathbf{\Omega},\mu)\to L_2(\mathbf{\Omega},\mu)} = \sup_{\|\phi\|_{L_2}\leq 1}|\int_{\mathbf{\Omega}}(K_1(\mathbf{x},\mathbf{y}) - K_2(\mathbf{x},\mathbf{y}))\phi(\mathbf{x})\phi(\mathbf{y})d\mu(\mathbf{x})d\mu(\mathbf{y})| \leq$$

$$\varepsilon\sup_{\|\phi\|_{L_2}\leq 1}\|\phi\|_{L_1(\mathbf{\Omega},\mu)}^2 \leq \varepsilon\sup_{\|\phi\|_{L_2}\leq 1}\|\phi\|_{L_2(\mathbf{\Omega},\mu)}^2 = \varepsilon.$$

Thus, $\|\mathrm{O}_{K_1} - \mathrm{O}_{K_2}\|_{L_2(\mathbf{\Omega},\mu)\to L_2(\mathbf{\Omega},\mu)} \leq \varepsilon$. The space of Hölder functions of order $\alpha \in (0,1)$ is denoted by $\Lambda^\alpha(\mathbb{R})$. For $f : \mathbb{R} \to \mathbb{R}$ from that space, we denote its $\alpha$-Hölder norm by $\|f\|_{\Lambda^\alpha(\mathbb{R})} = \sup_{x\neq y}\frac{|f(x)-f(y)|}{|x-y|^\alpha}$. According to a result of Aleksandrov and Peller [2016], for any Hilbert space $\mathcal{H}$ and bounded self-adjoint operators $A, B$ on $\mathcal{H}$, we have $\|f(A) - f(B)\|_{\mathcal{H}\to\mathcal{H}} \leq c(1-\alpha)^{-1}\|A-B\|_{\mathcal{H}\to\mathcal{H}}^\alpha$, where $c$ is a universal constant. For $f(x) = \sqrt{\max(x,0)}$ we have $\|f\|_{\Lambda^{1/2}(\mathbb{R})} = 1$ and, therefore, we have

$$\|\mathrm{O}_{K_1}^{1/2} - \mathrm{O}_{K_2}^{1/2}\|_{L_2(\mathbf{\Omega},\mu)\to L_2(\mathbf{\Omega},\mu)} \leq c\|\mathrm{O}_{K_1} - \mathrm{O}_{K_2}\|_{L_2(\mathbf{\Omega},\mu)\to L_2(\mathbf{\Omega},\mu)}^{1/2} \leq c\varepsilon^{1/2}.$$

$\square$

**Lemma 6.** *Let $\mu$ be a probabilistic nondegenerate Borel measure on compact $\mathbf{\Omega} \subseteq \mathbb{R}^n$. Let $K_1, K_2 : \mathbf{\Omega} \times \mathbf{\Omega} \to \mathbb{R}$ be two Mercer kernels such that $|K_1(\mathbf{x}, \mathbf{y}) - K_2(\mathbf{x}, \mathbf{y})| < \varepsilon$. Then, for any $f_1 \in \mathcal{H}_{K_1}$ there exists $f_2 \in \mathcal{H}_{K_2}$ such that $\|f_1\|_{\mathcal{H}_{K_1}} = \|f_2\|_{\mathcal{H}_{K_2}}$ and $\|f_1 - f_2\|_{L_2(\mathbf{\Omega}, \mu)} \leq c\varepsilon^{1/2}\|f_1\|_{\mathcal{H}_{K_1}}$.*

*Proof.* According to Corollary 4.13 from Cucker and Zhou [2007], the RKHS for the kernel $K : \mathbf{\Omega} \times \mathbf{\Omega} \to \mathbb{R}$ can be characterized as

$$\mathcal{H}_K = \mathrm{O}_K^{1/2}[L_2(\mathbf{\Omega}, \mu)],$$

and any function $f \in \mathcal{H}_K$ can be written as $f = \mathrm{O}_K^{1/2}[g], g \in L_2(\mathbf{\Omega}, \mu)$ with $\|f\|_{\mathcal{H}_K} = \|g\|_{L_2(\mathbf{\Omega}, \mu)}$. Thus, we have

$$B_{\mathcal{H}_{K_i}} = \{\mathrm{O}_{K_i}^{1/2}[\phi] \mid \phi \in L_2(\mathbf{\Omega}, \mu), \|\phi\|_{L_2} = 1\}.$$

Therefore, for $f_1 \in \mathcal{H}_{K_1}$ there exist $\phi \in L_2(\mathbf{\Omega}, \mu), \|\phi\|_{L_2(\mathbf{\Omega}, \mu)} = \|f_1\|_{\mathcal{H}_{K_1}}$ such that $f_1 = \mathrm{O}_{K_1}^{1/2}[\phi]$. Let us now define $f_2 = \mathrm{O}_{K_2}^{1/2}[\phi]$. By construction, $f_2 \in \mathcal{H}_{K_2}$ and $\|f_1\|_{\mathcal{H}_{K_1}} = \|f_2\|_{\mathcal{H}_{K_2}}$. Also, using Lemma 5, we have

$$\|f_1 - f_2\|_{L_2(\mathbf{\Omega}, \mu)} = \|(\mathrm{O}_{K_1}^{1/2} - \mathrm{O}_{K_2}^{1/2})[\phi]\|_{L_2(\mathbf{\Omega}, \mu)} \leq c\varepsilon^{1/2}\|\phi\|_{L_2(\mathbf{\Omega}, \mu)} = c\varepsilon^{1/2}\|f_1\|_{\mathcal{H}_{K_1}}.$$

$\square$

*Proof of Theorem 4.* First, Theorem 3 gives us

$$|\tilde{\Sigma}^{(L)}(\mathbf{x}, \mathbf{x}') - \Sigma^{(L)}(\mathbf{x}, \mathbf{x}')| \leq \varepsilon,$$

where

$$\varepsilon = R\|\sigma\|_\infty^4 \max(\|\sigma''''\|_\infty\|\sigma\|_\infty, \|\sigma'''\|_\infty\|\sigma'\|_\infty, \|\sigma''\|_\infty^2) \sum_{j=1}^{L-1} \frac{\max(2\|\sigma''\|_\infty\|\sigma\|_\infty, 2\|\sigma'\|_\infty^2, \frac{5}{2})^{2L-2j}(L-j)}{n_j}.$$

From Lemma 6 we conclude that there exists $f_1 \in \mathcal{H}_{\tilde{\Sigma}^{(L)}}$ such that $\|f_1\|_{\mathcal{H}_{\tilde{\Sigma}^{(L)}}} = \|f\|_{\mathcal{H}_{\Sigma^{(L)}}}$ and $\|f - f_1\|_{L_2(\mathbf{\Omega}, \mu)} \leq c\|f\|_{\mathcal{H}_{\Sigma^{(L)}}} \sqrt{\varepsilon}$.

Further, using Theorem 1, we construct $\tilde{f}(\mathbf{x}) = \sum_{i=1}^T w_i \sigma(W^{(i,L)} \sigma(\cdots \sigma(W^{(i,1)}\mathbf{x}) \cdots))$ such that $\|\tilde{f} - f_1\|_{L_2(\mathbf{\Omega}, \mu)} \leq \frac{\|\sigma\|_\infty\|f_1\|_{\mathcal{H}_{\tilde{\Sigma}^{(L)}}}}{\sqrt{T}} = \frac{\|\sigma\|_\infty\|f\|_{\mathcal{H}_{\Sigma^{(L)}}}}{\sqrt{T}}$. Finally, from the triangle inequality we conclude

$$\|\tilde{f} - f\|_{L_2(\mathbf{\Omega}, \mu)} \leq \|\tilde{f} - f_1\|_{L_2(\mathbf{\Omega}, \mu)} + \|f_1 - f\|_{L_2(\mathbf{\Omega}, \mu)} \leq \frac{\|\sigma\|_\infty\|f\|_{\mathcal{H}_{\Sigma^{(L)}}}}{\sqrt{T}} + c\|f\|_{\mathcal{H}_{\Sigma^{(L)}}} \sqrt{\varepsilon}.$$

$\square$

# E PROPERTIES OF $\overline{\sigma}$

The following lemma is a direct generalization of Lemma 12 from Daniely et al. [2016]. We give its proof for completeness.

**Lemma 7.** *Suppose that $\phi \in C^2(\mathbb{R}^2)$ and $\phi(\mathbf{z})$ decays faster than $e^{-\gamma\|\mathbf{z}\|^2}$ for any $\gamma > 0$ (as $\|\mathbf{z}\| \to +\infty$). Let $\Phi : \mathcal{M}_+ \to \mathbb{R}$ be defined by $\Phi(\Sigma) = \mathbb{E}_{(X,Y)\sim\mathcal{N}(\mathbf{0}, \Sigma)}[\phi(X, Y)]$. Then, $\Phi \in C^1(\mathcal{M}_+)$ and*

$$\frac{\partial \Phi(\Sigma)}{\partial \Sigma} = \frac{1}{2}\mathbb{E}_{(X,Y)\sim\mathcal{N}(\mathbf{0}, \Sigma)}[\frac{\partial^2 \phi}{\partial^2 \mathbf{z}}].$$

*Proof.* By definition, we have

$$\Phi(\Sigma) = \frac{1}{2\pi\sqrt{\det(\Sigma)}} \int_{\mathbb{R}^2} \phi(\mathbf{z}) e^{-\frac{\mathbf{z}^\top \Sigma^{-1} \mathbf{z}}{2}} d\mathbf{z}.$$

Let us denote $\Sigma = \begin{bmatrix} \Sigma_{11} & \Sigma_{12} \\ \Sigma_{21} & \Sigma_{22} \end{bmatrix}$. It is well-known that for symmetric matrices we have $\frac{\partial(\det(\Sigma))}{\partial \Sigma} = \frac{\partial(\Sigma_{11}\Sigma_{22} - \Sigma_{12}\Sigma_{21})}{\partial \Sigma} = \begin{bmatrix} \Sigma_{22} & -\Sigma_{21} \\ -\Sigma_{12} & \Sigma_{11} \end{bmatrix} = \det(\Sigma)\Sigma^{-1}$. Let $\mathbf{z} = [u, v]^\top$ and $\mathrm{adj}(\Sigma)$ be the adjoint matrix of $\Sigma$. Also, symmetricity of $\Sigma$ gives us $\frac{\partial(\mathbf{z}^\top \Sigma^{-1} \mathbf{z})}{\partial \Sigma} = -\Sigma^{-1}\mathbf{z}\mathbf{z}^\top \Sigma^{-1}$, due to

$$\frac{\partial(\mathbf{z}^\top \Sigma^{-1} \mathbf{z})}{\partial \Sigma} = \frac{\partial}{\partial \Sigma}\left(\frac{\mathbf{z}^\top \mathrm{adj}(\Sigma)\mathbf{z}}{\det(\Sigma)}\right) =$$

$$-\frac{1}{\det(\Sigma)^2}\begin{bmatrix} \Sigma_{22} & -\Sigma_{21} \\ -\Sigma_{12} & \Sigma_{11} \end{bmatrix}(\Sigma_{22}u^2 + \Sigma_{11}v^2 - \Sigma_{12}uv - \Sigma_{21}uv) + \det(\Sigma)^{-1}\begin{bmatrix} v^2 & -uv \\ -uv & u^2 \end{bmatrix} =$$

$$\det(\Sigma)^{-2}\begin{bmatrix} -\Sigma_{22}^2 u^2 - \Sigma_{12}\Sigma_{21}v^2 + \Sigma_{22}\Sigma_{12}uv + \Sigma_{22}\Sigma_{21}uv & \Sigma_{21}\Sigma_{22}u^2 + \Sigma_{21}\Sigma_{11}v^2 - (\Sigma_{11}\Sigma_{22} + \Sigma_{21}^2)uv \\ \Sigma_{12}\Sigma_{22}u^2 + \Sigma_{12}\Sigma_{11}v^2 - (\Sigma_{11}\Sigma_{22} + \Sigma_{12}^2)uv & -\Sigma_{12}\Sigma_{21}u^2 - \Sigma_{11}^2 v^2 + \Sigma_{11}\Sigma_{12}uv + \Sigma_{11}\Sigma_{21}uv \end{bmatrix} =$$

$$-\det(\Sigma)^{-2}\begin{bmatrix} (\Sigma_{22}u - \Sigma_{21}v)^2 & (\Sigma_{22}u - \Sigma_{21}v)(\Sigma_{11}v - \Sigma_{21}u) \\ (\Sigma_{22}u - \Sigma_{21}v)(\Sigma_{11}v - \Sigma_{21}u) & (\Sigma_{11}v - \Sigma_{21}u)^2 \end{bmatrix} = -\Sigma^{-1}\mathbf{z}\mathbf{z}^\top \Sigma^{-1}.$$

Therefore,

$$\frac{\partial \Phi}{\partial \Sigma} = \frac{1}{2\pi}\int_{\mathbb{R}^2}\phi(\mathbf{z})(-\frac{1}{2}\det(\Sigma)^{-\frac{3}{2}}\det(\Sigma)\Sigma^{-1} + \frac{1}{2}\det(\Sigma)^{-\frac{1}{2}}\Sigma^{-1}\mathbf{z}\mathbf{z}^\top \Sigma^{-1})e^{-\frac{\mathbf{z}^\top \Sigma^{-1}\mathbf{z}}{2}}\,d\mathbf{z} =$$

$$-\frac{1}{2\pi\sqrt{\det(\Sigma)}}\int_{\mathbb{R}^2}\phi(\mathbf{z})\frac{1}{2}(\Sigma^{-1} - \Sigma^{-1}\mathbf{z}\mathbf{z}^\top \Sigma^{-1})e^{-\frac{\mathbf{z}^\top \Sigma^{-1}\mathbf{z}}{2}}\,d\mathbf{z}.$$

Since

$$\frac{\partial}{\partial \mathbf{z}}(e^{-\frac{\mathbf{z}^\top \Sigma^{-1}\mathbf{z}}{2}}) = -(\Sigma^{-1}\mathbf{z})^\top e^{-\frac{\mathbf{z}^\top \Sigma^{-1}\mathbf{z}}{2}},$$

$$\frac{\partial^2}{\partial \mathbf{z}^2}(e^{-\frac{\mathbf{z}^\top \Sigma^{-1}\mathbf{z}}{2}}) = (\Sigma^{-1}\mathbf{z}\mathbf{z}^\top \Sigma^{-1} - \Sigma^{-1})e^{-\frac{\mathbf{z}^\top \Sigma^{-1}\mathbf{z}}{2}},$$

we conclude

$$\frac{\partial \Phi}{\partial \Sigma} = \frac{1}{2}\frac{1}{2\pi\sqrt{\det(\Sigma)}}\int_{\mathbb{R}^2}\phi(\mathbf{z})\frac{\partial^2}{\partial \mathbf{z}^2}(e^{-\frac{\mathbf{z}^\top \Sigma^{-1}\mathbf{z}}{2}})\,d\mathbf{z} = \frac{1}{2}\mathbb{E}_{(X,Y)\sim\mathcal{N}(\mathbf{0},\Sigma)}[\frac{\partial^2\phi}{\partial \mathbf{z}^2}].$$

$\square$

From the previous lemma the following result is straightforward.

**Lemma 8.** *For any $\Sigma \in \mathcal{M}_+$, we have*

$$|\frac{\partial \overline{\sigma}(\Sigma)}{\partial \Sigma_{a,b}}| \leq \frac{1}{2}\max(\|\sigma\|_\infty\|\sigma''\|_\infty, \|\sigma'\|_\infty^2),$$

*and*

$$|\frac{\partial^2\overline{\sigma}(\Sigma)}{\partial \Sigma_{a,b}\partial \Sigma_{c,d}}| \leq \frac{1}{4}\max(\|\sigma''''\|_\infty\|\sigma\|_\infty, \|\sigma'''\|_\infty\|\sigma'\|_\infty, \|\sigma''\|_\infty^2).$$

*Proof.* From the previous lemma we have $|\frac{\partial \overline{\sigma}(\Sigma)}{\partial \Sigma_{1,1}}| = \frac{1}{2}|\mathbb{E}_{(u,v)\sim\mathcal{N}(\mathbf{0},\Sigma)}[\sigma''(u)\sigma(v)]| \leq \frac{1}{2}\|\sigma''\|_\infty\|\sigma\|_\infty$. Analogously, $|\frac{\partial \overline{\sigma}(\Sigma)}{\partial \Sigma_{2,2}}| \leq \frac{1}{2}\|\sigma''\|_\infty\|\sigma\|_\infty$. For cross terms we have $|\frac{\partial \overline{\sigma}(\Sigma)}{\partial \Sigma_{1,2}}| = \frac{1}{2}|\mathbb{E}_{(u,v)\sim\mathcal{N}(\mathbf{0},\Sigma)}[\sigma'(u)\sigma'(v)]| \leq \frac{1}{2}\|\sigma'\|_\infty^2$.

For second derivatives using the previous lemma twice gives us $\frac{\partial^2\overline{\sigma}(\Sigma)}{\partial \Sigma_{a,b}\partial \Sigma_{c,d}} = \frac{1}{4}\mathbb{E}_{(u,v)\sim\mathcal{N}(\mathbf{0},\Sigma)}[\frac{\partial^4}{\partial z_a\partial z_b\partial z_c\partial z_d}(\sigma(u)\sigma(v))]$ where $z_1 = u, z_2 = v$. Therefore, $|\frac{\partial^2\overline{\sigma}(\Sigma)}{\partial \Sigma_{a,b}\partial \Sigma_{c,d}}| \leq \frac{1}{4}\max(\|\sigma''''\|_\infty\|\sigma\|_\infty, \|\sigma'''\|_\infty\|\sigma'\|_\infty, \|\sigma''\|_\infty^2)$. $\square$

# F PROOF OF THEOREM 5

*Proof.* From the Funk-Hecke formula we obtain that for $\mathbf{x}, \mathbf{y} \in \mathbb{R}^n$,

$$e^{-\mathrm{i}\|\mathbf{x}\|\|\mathbf{y}\|\widehat{\mathbf{x}}^T\widehat{\mathbf{y}}} = \sum_{k=0}^{\infty} \mu_k(\|\mathbf{x}\|\|\mathbf{y}\|) \sum_{j=1}^{N(n,k)} Y_{k,j}(\widehat{\mathbf{x}})Y_{k,j}(\widehat{\mathbf{y}}),$$

where $\widehat{\mathbf{x}} = \frac{\mathbf{x}}{\|\mathbf{x}\|}$,

$$\mu_k(r) = \frac{\Gamma(\frac{n}{2})}{\sqrt{\pi}\Gamma(\frac{n-1}{2})} \int_{-1}^{1} e^{-\mathrm{i}rt}P_k(t)(1-t^2)^{\frac{n-3}{2}}dt,$$

and $P_k(x) = \frac{(-1)^k\Gamma(\frac{n-1}{2})}{2^k\Gamma(k+\frac{n-1}{2})}(1-t^2)^{-\frac{n-3}{2}}\frac{d^k}{dt^k}[(1-t^2)^{k+\frac{n-3}{2}}]$ is the $k$th Gegenbauer polynomial of the parameter $\alpha = \frac{n-2}{2}$ (Rodrigues' formula is derived in Frye and Efthimiou [2012]). Using integration by parts we obtain

$$\mu_k(r) \propto^n \frac{(-1)^k}{2^k\Gamma(k+\frac{n-1}{2})}(-\mathrm{i}r)^k \int_{-1}^{1} e^{-\mathrm{i}rt}(1-t^2)^{k+\frac{n-3}{2}}dt \propto^n$$

$$\frac{(-1)^k}{2^k\Gamma(k+\frac{n-1}{2})}(-\mathrm{i}r)^k \frac{\Gamma(k+\frac{n-1}{2})J_{k+\frac{n-2}{2}}(r)}{(r/2)^{k+\frac{n-2}{2}}} \propto^n \frac{\mathrm{i}^k J_{k+\frac{n-2}{2}}(r)}{r^{\frac{n-2}{2}}},$$

where $J_k$ is the Bessel function of order $k$. In the latter, we used the Formula 8.411.10 from Gradshteyn and Ryzhik [2015].

Thus, we obtain the plain wave expansion in $\mathbb{R}^n$:

$$e^{-\mathrm{i}\mathbf{x}^T\mathbf{y}} \propto^n \sum_{k=0}^{\infty} \frac{\mathrm{i}^k}{\|\mathbf{x}\|^{\frac{n-2}{2}}\|\mathbf{y}\|^{\frac{n-2}{2}}} J_{k+\frac{n-2}{2}}(\|\mathbf{x}\|\|\mathbf{y}\|) \sum_{j=1}^{N(n,k)} Y_{k,j}(\widehat{\mathbf{x}})Y_{k,j}(\widehat{\mathbf{y}}).$$

Note that our version of the plain wave expansion formula slightly differs from the one in which spherical harmonics are complex-valued (e.g. see page 48 of Avery and Avery [2018]).

Further, our goal will be to construct a function $f : \mathbb{S}^{n-1} \to \mathbb{R}$ that has a large norm $C_{f,\mathbb{S}^{n-1}}$, yet a moderate norm in $\mathcal{H}_K$. We will use the following key lemma.

**Lemma 9.** *There exists* $\mathbf{z} \in \mathbb{R}^{N(n,k)}$ *such that* $\sum_{j=1}^{N(n,k)} z_j^2 = 1$ *and*

$$\|\sum_{j=1}^{N(n,k)} z_j Y_{k,j}\|_{L_\infty(\mathbb{S}^{n-1})} \ll^n \sqrt{\log N(n,k)}.$$

As can be seen from the proof below, the vector $\mathbf{z}$ can be simply generated according to the uniform distribution on $\mathbb{S}^{n-1}$.

*Proof.* Let us define a new norm $\|\cdot\|_{(\infty)}$ on $\mathbb{R}^{N(n,k)}$ by

$$\|\boldsymbol{\xi}\|_{(\infty)} = \|\sum_{j=1}^{N(n,k)} \xi_j Y_{k,j}\|_{L_\infty(\mathbb{S}^{n-1})}.$$

The Levy mean of the norm $\|\cdot\|_{(\infty)}$ is defined by

$$M(\|\cdot\|_{(\infty)}) = \Big(\int_{\mathbb{S}^{n-1}} \|\boldsymbol{\xi}\|_{(\infty)}^2 d\nu(\boldsymbol{\xi})\Big)^{1/2}.$$

From Theorem 1 of Kushpel and Tozoni [2012] we have

$$M(\|\cdot\|_{(\infty)}) \leq C \log^{1/2} N(n,k).$$

where $C$ is some universal constant, or $\int_{\mathbb{S}^{n-1}} \|\boldsymbol{\xi}\|_{(\infty)}^2 d\nu(\boldsymbol{\xi}) \ll \log N(n,k)$. Therefore, there exists $\mathbf{z} \in \mathbb{R}^{N(n,k)}$ such that $\|\mathbf{z}\|_{(\infty)}^2 \ll \omega_{n-1}^{-1} \log N(n,k)$ and this completes the proof. $\square$

Let $\sigma_k = \sqrt{\lambda_k}$. Let us denote $Y_k = \sum_{j=1}^{N(n,k)} z_j Y_{k,j}$ where $\mathbf{z}$ satisfies the condition from the previous lemma. Since $\|\sigma_k Y_k\|_{\mathcal{H}_K} = 1$ we will be interested in the norm of $\sigma_k Y_k$ in the Barron space.

Let $g : \mathbb{R}^n \to \mathbb{R}$ be such that $g|_{\mathbb{S}^{n-1}} = \sigma_k Y_k$, $\int_{\mathbb{R}^n} \|\boldsymbol{\omega}\| |\widehat{g}(\boldsymbol{\omega})| d\boldsymbol{\omega} < +\infty$ and

$$g(\mathbf{x}) = g(\mathbf{0}) + \int_{\mathbb{R}^n} (e^{\mathrm{i}\boldsymbol{\omega}^\top \mathbf{x}} - 1)\widehat{g}(\boldsymbol{\omega}) d\boldsymbol{\omega}.$$

Then, Fubini's theorem combined with the plane wave expansion gives us

$$\sigma_k = \int_{\mathbb{S}^{n-1}} g(\mathbf{x}) Y_k(\mathbf{x}) d\nu(\mathbf{x}) = \int_{\mathbb{R}^n} \widehat{g}(\boldsymbol{\omega}) \int_{\mathbb{S}^{n-1}} e^{\mathrm{i}\boldsymbol{\omega}^\top \mathbf{x}} Y_k(\mathbf{x}) d\nu(\mathbf{x}) d\boldsymbol{\omega} \propto^n \int_{\mathbb{R}^n} \widehat{g}(\boldsymbol{\omega}) \frac{(-\mathrm{i})^k}{\|\boldsymbol{\omega}\|^{\frac{n-2}{2}}} J_{k+\frac{n-2}{2}}(\|\boldsymbol{\omega}\|) Y_k(\widehat{\boldsymbol{\omega}}) d\boldsymbol{\omega}.$$

By construction, we have $|Y_k(\widehat{\boldsymbol{\omega}})| \ll^n \sqrt{\log N(n,k)}$. After using the Hölder's inequality, we plug in the latter inequality into the former and obtain

$$\sigma_k \leq^n \sqrt{\log N(n,k)} \int_{\mathbb{R}^n} \|\boldsymbol{\omega}\| |\widehat{g}(\boldsymbol{\omega})| d\boldsymbol{\omega} \sup_{\boldsymbol{\omega} \in \mathbb{R}^n} \frac{|J_{k+\frac{n-2}{2}}(\|\boldsymbol{\omega}\|)|}{\|\boldsymbol{\omega}\|^{\frac{n}{2}}}.$$

Let us fix $0 < \gamma < 1$. For the Bessel function $J_\nu, \nu = k + \frac{n-2}{2}$ we have the Meissel's formula (see page 227 in Watson [1980], also see an equivalent Formula 8.452 from Gradshteyn and Ryzhik [2015]),

$$J_\nu(\nu z) \asymp \frac{(\nu z)^\nu e^{\nu \sqrt{1-z^2}}}{e^\nu \Gamma(\nu+1)(1-z^2)^{\frac{1}{4}}(1+\sqrt{1-z^2})^\nu},$$

which holds for any $z \in [0, \gamma]$ and a large $\nu$. Therefore,

$$\max_{r \in [0, \nu\gamma]} r^{-\frac{n}{2}} J_\nu(r) = \max_{z \in [0, \gamma]} (\nu z)^{-\frac{n}{2}} J_\nu(\nu z) \ll \max_{z \in [0, \gamma]} \frac{(\nu z)^{\nu - \frac{n}{2}} e^{\nu \sqrt{1-z^2}}}{e^\nu \Gamma(\nu+1)(1-z^2)^{\frac{1}{4}}(1+\sqrt{1-z^2})^\nu}.$$

A derivative of $(\nu - \frac{n}{2}) \log z + \nu\sqrt{1-z^2} - \nu \log(1+\sqrt{1-z^2}) - \frac{1}{4}\log(1-z^2)$ is $z(\frac{\nu-\frac{n}{2}}{z^2} - \frac{\nu}{1+\sqrt{1-z^2}} + \frac{1}{2-2z^2})$. The function $\frac{\nu-\frac{n}{2}}{t} + \frac{1}{2-2t}$ attains its minimum in $[0,1]$ when $(\frac{\nu-\frac{n}{2}}{t} + \frac{1}{2-2t})' = -\frac{\nu-\frac{n}{2}}{t^2} + \frac{1}{2(1-t)^2} = 0$, i.e. when $t = ((2\nu-n)^{-1/2}+1)^{-1}$. For large $\nu$ we have $\frac{\nu-\frac{n}{2}}{((2\nu-n)^{-1/2}+1)^{-1}} = \nu - \frac{n}{2} + \frac{1}{\sqrt{2}}\sqrt{\nu - \frac{n}{2}} \geq \nu$. Thus, if $\nu$ is sufficiently large we always have

$$\frac{\nu-\frac{n}{2}}{t} + \frac{1}{2-2t} \geq \nu \geq \frac{\nu}{1+\sqrt{1-t}} \quad \forall t \in [0,1].$$

Therefore, the RHS of Meissel's formula is a growing function of $z$ and the maximum is attained at $z = \gamma$. In other words, we reduced the maximization of $r^{-\frac{n}{2}} J_\nu(r)$ over $[0, +\infty)$ to the maximization over $[\nu\gamma, +\infty)$. Using a uniform bound from Krasikov [2006], i.e. $J_\nu(r) \ll \nu^{-\frac{1}{3}}$, we conclude that $\max_{r \in [\nu\gamma, +\infty]} r^{-\frac{n}{2}} J_\nu(r) \ll^n \nu^{-\frac{n}{2}-\frac{1}{3}}$.

Thus, we have

$$\sigma_k \leq^n \sqrt{\log N(n,k)} \nu^{-\frac{n}{2}-\frac{1}{3}} \int_{\mathbb{R}^n} \|\boldsymbol{\omega}\| |\widehat{g}(\boldsymbol{\omega})| d\boldsymbol{\omega}.$$

The latter directly gives a lower bound $\frac{\sigma_k k^{\frac{n}{2}+\frac{1}{3}}}{\sqrt{\log N(n,k)}}$ on the norm of $\sigma_k Y_k \in \mathcal{H}_K$ in the Barron's space of $\mathbb{S}^{n-1}$ and completes the proof. $\qquad\square$

**Remark 4.** *The lower bound on $C_{\sigma_k Y_k, \mathbb{S}^{n-1}}$ that was given in the latter proof can be turned into a lower bound on $C_{Y_k, \mathbb{S}^{n-1}}$, that is $C_{Y_k, \mathbb{S}^{n-1}} \geq^n \frac{k^{\frac{n}{2}+\frac{1}{3}}}{\sqrt{\log N(n,k)}}$. Since $\|\sigma_k Y_k\|_{\mathcal{H}_K} = 1$ we also have $\|Y_k\|_{\mathcal{H}_K} = \sigma_k^{-1}$. Thus, for most of popular activation functions, both $C_{Y_k, \mathbb{S}^{n-1}}$ and $\|Y_k\|_{\mathcal{H}_K}$ grow quite rapidly with $k$. This property makes $Y_k$ a target function for testing boundaries of our approximation theory, Barron's theorem, and the NTK theory.*

# G  PROOF SKETCH OF THEOREM 6

Any function $f \in B_{\mathcal{H}_K}$ can be represented as

$$f = \sum_{k=0}^{\infty} \sigma_k x_k Z_k$$

where $\sum_{k=0}^{\infty} x_k^2 \leq 1$ and $Z_k : \mathbb{S}^{n-1} \to \mathbb{R}$ is a spherical harmonics of order $k$ such that $\|Z_k\|_{L_2(\mathbb{S}^{n-1})} = 1$. Our goal is to construct $g : \mathbb{R}^n \to \mathbb{R}$ such that $g|_{\mathbb{S}^{n-1}} = f$ and the integral $\int_{\mathbb{R}^n} \|\boldsymbol{\omega}\| |\widehat{g}(\boldsymbol{\omega})| d\boldsymbol{\omega}$ is as small as possible. First we will define $g_k$ such that $g_k|_{\mathbb{S}^{n-1}} = \sigma_k Z_k$ and then set $g = \sum_{k=0}^{\infty} x_k g_k$. The key inequality that bounds the latter integral is the following one:

$$\int_{\mathbb{R}^n} \|\boldsymbol{\omega}\| |\widehat{g}(\boldsymbol{\omega})| d\boldsymbol{\omega} \leq \sum_{k=0}^{\infty} |x_k| \int_{\mathbb{R}^n} \|\boldsymbol{\omega}\| |\widehat{g_k}(\boldsymbol{\omega})| d\boldsymbol{\omega} \leq \Big( \sum_{k=0}^{\infty} \big( \int_{\mathbb{R}^n} \|\boldsymbol{\omega}\| |\widehat{g_k}(\boldsymbol{\omega})| d\boldsymbol{\omega} \big)^2 \Big)^{1/2}.$$

Thus, we only need the series $\sum_{k=0}^{\infty} (\int_{\mathbb{R}^n} \|\boldsymbol{\omega}\| |\widehat{g_k}(\boldsymbol{\omega})| d\boldsymbol{\omega})^2$ to be converging and this will guarantee that $B_{\mathcal{H}_K}$ is bounded in the Barron space.

First, let us define $G_k$ in such a way that

$$\widehat{G_k}(\boldsymbol{\omega}) = \sigma_k t_k(\|\boldsymbol{\omega}\|) \delta(\|\boldsymbol{\omega}\| - (k + \frac{n-2}{2})) Z_k(\widehat{\boldsymbol{\omega}}),$$

where $t_k$ is to be specified later in order to satisfy $G_k|_{\mathbb{S}^{n-1}} = \sigma_k Z_k$ and $\delta$ is the Dirac delta function. Note that $\widehat{G_k}$ is a tempered distribution, not an ordinary function. Thus, $G_k$ equals

$$G_k(\mathbf{x}) = \int_{\mathbb{R}^n} \widehat{G_k}(\boldsymbol{\omega}) e^{i\boldsymbol{\omega}^\top \mathbf{x}} d\boldsymbol{\omega} = \int_{\mathbb{R}^n} \widehat{G_k}(\boldsymbol{\omega}) \sum_{k'=0}^{\infty} \frac{(-i)^{k'}}{\|\mathbf{x}\|^{\frac{n-2}{2}} \|\boldsymbol{\omega}\|^{\frac{n-2}{2}}} J_{k' + \frac{n-2}{2}}(\|\mathbf{x}\| \|\boldsymbol{\omega}\|) \sum_{j=1}^{N(n,k')} Y_{k',j}(\widehat{\mathbf{x}}) Y_{k',j}(\widehat{\boldsymbol{\omega}}) d\boldsymbol{\omega} =$$

$$\frac{(-i)^k \sigma_k Z_k(\widehat{\mathbf{x}})}{\|\mathbf{x}\|^{\frac{n-2}{2}}} \int_0^{\infty} t_k(r) \delta(r - (k + \frac{n-2}{2})) J_{k + \frac{n-2}{2}}(\|\mathbf{x}\| r) r^{\frac{n}{2}} dr.$$

If $\mathbf{x} \in \mathbb{S}^{n-1}$, then $G_k(\mathbf{x}) = (-i)^k \sigma_k Z_k(\mathbf{x}) t_k(k + \frac{n-2}{2}) J_{k + \frac{n-2}{2}}(k + \frac{n-2}{2})(k + \frac{n-2}{2})^{\frac{n}{2}}$. Thus, in order to have $G_k(\mathbf{x}) = \sigma_k Z_k(\mathbf{x})$ we need to set $t_k$ to any smooth function such that $t_k(k + \frac{n-2}{2}) = \frac{i^k}{J_{k + \frac{n-2}{2}}(k + \frac{n-2}{2})(k + \frac{n-2}{2})^{\frac{n}{2}}}$. Since $J_{k + \frac{n-2}{2}}(k + \frac{n-2}{2}) \asymp (k + \frac{n-2}{2})^{-1/3}$, we conclude that

$$\Big| t(k + \frac{n-2}{2}) \Big| \ll^n k^{-\frac{n}{2} + \frac{1}{3}}.$$

Therefore,

$$\int_{\mathbb{R}^n} \|\boldsymbol{\omega}\| |\widehat{G_k}(\boldsymbol{\omega})| d\boldsymbol{\omega} = \sigma_k \int_0^{\infty} t_k(r) r^n \delta(r - (k + \frac{n-2}{2})) dr \int_{\mathbb{S}^{n-1}} |Z_k(\widehat{\boldsymbol{\omega}})| d\nu(\widehat{\boldsymbol{\omega}}) \ll^n$$

$$\sigma_k t_k(k + \frac{n-2}{2})(k + \frac{n-2}{2})^n \ll^n \sigma_k k^{\frac{n}{2} + \frac{1}{3}}.$$

Now it remains to define $g_k$ in such a way that $\widehat{g_k}$ is an ordinary function, unlike $\widehat{G_k}$. This can be done by simply substituting the delta function with the Gaussian $N_\varepsilon(x) = (2\pi\epsilon^2)^{-\frac{1}{2}} e^{-\frac{x^2}{2\epsilon^2}}$, i.e. by setting

$$\widehat{g_k}(\boldsymbol{\omega}) = \sigma_k t_k(\|\boldsymbol{\omega}\|) N_{\varepsilon_k}(\|\boldsymbol{\omega}\| - (k + \frac{n-2}{2})) Z_k(\widehat{\boldsymbol{\omega}}),$$

for the sequence $\varepsilon_k = 2^{-2^k}$. After that, both $g_k$ and $\widehat{g_k}$ are ordinary functions and this completes the proof.

# H   THE CASE OF THE GAUSSIAN ACTIVATION FUNCTION

Let us now consider the case $\sigma(x) = e^{-\frac{x^2}{2}}$. For this case, the NNGP can be computed directly. By definition, we have

$$K(\mathbf{x}, \mathbf{x}') = \int_{\mathbb{R}^2} \sigma(u)\sigma(v)G(u, v|\Sigma_{\mathbf{x},\mathbf{x}'})dudv,$$

where $G(\mathbf{s}|\Sigma_{\mathbf{x},\mathbf{x}'}) = \frac{1}{2\pi \det(\Sigma_{\mathbf{x},\mathbf{x}'})^{1/2}}\exp(-\frac{\mathbf{s}^\top \Sigma_{\mathbf{x},\mathbf{x}'}^{-1}\mathbf{s}}{2})$. Therefore,

$$\frac{1}{2\pi}\int_{\mathbb{R}^2} e^{-\frac{\|\mathbf{s}\|^2}{2}}\frac{1}{\det(\Sigma_{\mathbf{x},\mathbf{x}'})^{1/2}}\exp(-\frac{\mathbf{s}^\top \Sigma_{\mathbf{x},\mathbf{x}'}^{-1}\mathbf{s}}{2})d\mathbf{s} = \frac{1}{\det(I_2 + \Sigma_{\mathbf{x},\mathbf{x}'})^{1/2}}.$$

Since $\det(I_2 + \Sigma_{\mathbf{x},\mathbf{x}'}) = 1 + \mathbf{x}^\top\mathbf{x} + \mathbf{x}'^\top\mathbf{x}' + \|\mathbf{x}\|^2 \cdot \|\mathbf{x}'\|^2 - (\mathbf{x}^\top\mathbf{x}')^2$, we conclude

$$K(\mathbf{x}, \mathbf{x}') = \frac{1}{(1 + \mathbf{x}^\top\mathbf{x} + \mathbf{x}'^\top\mathbf{x}' + \|\mathbf{x}\|^2 \cdot \|\mathbf{x}'\|^2 - (\mathbf{x}^\top\mathbf{x}')^2)^{1/2}}. \tag{9}$$

Let us analyze further the case $\mathbf{\Omega} = \mathbb{S}^{n-1}$.

*Proof of Theorem 7.*  For that case we have

$$K(\mathbf{x}, \mathbf{x}') = \frac{1}{(4 - (\mathbf{x}^\top\mathbf{x}')^2)^{1/2}} = f(\mathbf{x}^\top\mathbf{x}'),$$

where $f(t) = \frac{1}{\sqrt{4-t^2}}$.

From the Funk-Hecke formula we obtain

$$\lambda_k = \frac{\Gamma(\frac{n}{2})}{\sqrt{\pi}\Gamma(\frac{n-1}{2})}\int_{-1}^{1} f(t)P_k(t)(1-t^2)^{\frac{n-3}{2}}dt$$

where $P_k$ is the $k$th Gegenbauer polynomial of the parameter $\alpha = \frac{n-2}{2}$. Since $P_{2k+1}$ is an odd function, we conclude that $\lambda_{2k+1} = 0$. Let us now concentrate on the calculation of $\lambda_{2k}$.

For $t \in [-1, 1]$ we have

$$f(t) = \frac{1}{(4-t^2)^{1/2}} = \frac{1}{2}\sum_{i=0}^{\infty}(-1)^i\binom{-1/2}{i}(\frac{t}{2})^{2i} = \frac{1}{2}\sum_{i=0}^{\infty}\frac{\binom{2i}{i}}{2^{4i}}t^{2i} \Rightarrow$$

$$\int_{-1}^{1} f(t)P_{2k}(t)(1-t^2)^{\frac{n-3}{2}}dt = \sum_{i=0}^{\infty}\frac{\binom{2i}{i}}{2^{4i}}a_i^k,$$

where $a_i^k = \int_0^1 t^{2i}P_{2k}(t)(1-t^2)^{\frac{n-3}{2}}dt$. Note that $a_i^k = 0$ for $k > i$ due to the fact that $P_{2k}$ is orthogonal to $x^{2i}$. Using Rodrigues' formula we conclude

$$a_i^k = \frac{\Gamma(\frac{n-1}{2})}{2^{2k}\Gamma(2k + \frac{n-1}{2})}\int_0^1 t^{2i}\frac{d}{dt^{2k}}[(1-t^2)^{2k+\frac{n-3}{2}}]dt.$$

The expression $\int_0^1 t^{2i}\frac{d}{dt^{2k}}[(1-t^2)^{2k+\frac{n-3}{2}}]dt$ is nonzero for $i \geq k$ and integration by parts gives us

$$\int_0^1 t^{2i}\frac{d}{dt^{2k}}[(1-t^2)^{2k+\frac{n-3}{2}}]dt = \frac{(2i)!}{(2i-2k)!}\int_0^1 t^{2i-2k}(1-t^2)^{2k+\frac{n-3}{2}}dt =$$

$$\frac{(2i)!}{(2i-2k)!}\int_0^1 u^{2k+\frac{n-3}{2}}(1-u)^{i-k}\frac{du}{2\sqrt{1-u}} = \frac{(2i)!}{2(2i-2k)!}\frac{\Gamma(2k+\frac{n-1}{2})\Gamma(i-k+\frac{1}{2})}{\Gamma(k+i+\frac{n}{2})}.$$

Thus,

$$a_i^k = \frac{\Gamma(\frac{n-1}{2})}{2^{2k+1}}\frac{(2i)!\Gamma(i-k+\frac{1}{2})}{(2i-2k)!\Gamma(k+i+\frac{n}{2})},$$

and, therefore,

$$\int_{-1}^{1} f(t) P_{2k}(t)(1-t^2)^{\frac{n-3}{2}} dt = \frac{\Gamma(\frac{n-1}{2})}{2^{2k+1}} \sum_{i=k}^{\infty} \frac{\binom{2i}{i}}{2^{4i}} \frac{(2i)! \Gamma(i-k+\frac{1}{2})}{(2i-2k)! \Gamma(k+i+\frac{n}{2})}.$$

Using Stirling's formula, the first term in the latter sum can be bounded by $\frac{\binom{2k}{k}}{2^{4k}} \frac{(2k)! \Gamma(\frac{1}{2})}{\Gamma(2k+\frac{n}{2})} \ll \frac{2^{-2k}}{k^{1/2}} (2k)^{-\frac{n}{2}+1} \ll_n k^{-\frac{n}{2}}$. For terms starting from the second, we can apply Stirling's formula for $(2i-2k)!$ also:

$$\frac{\binom{2i}{i}(2i)!}{2^{4i}} \frac{\Gamma(i-k+\frac{1}{2})}{(2i-2k)! \Gamma(k+i+\frac{n}{2})} \asymp$$

$$\frac{(2i)^{4i+1}e^{-4i}}{2^{4i}i^{2i+1}e^{-2i}} \frac{1}{(2i-2k)^{2i-2k+\frac{1}{2}}e^{-(2i-2k)}} \frac{(i-k-\frac{1}{2})^{i-k}e^{-(i-k)}}{(k+i+\frac{n}{2}-1)^{k+i+\frac{n-1}{2}}e^{-(k+i+\frac{n}{2})}} \ll$$

$$2^{-(2i-2k)}e^{\frac{n}{2}} \frac{i^{2i}}{(i-k)^{i-k+\frac{1}{2}}(k+i+\frac{n}{2}-1)^{k+i+\frac{n-1}{2}}}.$$

Since $(x+\frac{1}{2})\log x$ is convex for $x > \frac{1}{2}$, we conclude that

$$2(i+\frac{n}{4})\log(i+\frac{n-2}{4}) \leq (i-k+\frac{1}{2})\log(i-k) + (k+i+\frac{n-1}{2})\log(k+i+\frac{n}{2}-1).$$

Therefore, we can proceed by bounding the previous expression with

$$2^{-(2i-2k)}e^{\frac{n}{2}} \frac{i^{2i}}{(i+\frac{n-2}{4})^{2(i+\frac{n}{4})}} \ll 2^{-(2i-2k)}i^{-\frac{n}{2}}.$$

Thus, we obtained $2^{-(2k+1)} \sum_{i=k}^{\infty} \frac{\binom{2i}{i}}{2^{4i}} \frac{(2i)! \Gamma(i-k+\frac{1}{2})}{(2i-2k)! \Gamma(k+i+\frac{n}{2})} \ll_n \sum_{i=k}^{\infty} 2^{-2i}i^{-\frac{n}{2}}$ and this leads to the final conclusion

$$\lambda_{2k} \ll_n \int_{k}^{\infty} 2^{-2x}x^{-\frac{n}{2}} dx \ll 2^{-2k}k^{-\frac{n}{2}}.$$

$\square$

# I  THE CASE OF THE COSINE AND THE SINE ACTIVATION FUNCTIONS

Since we could not find the derivation of the NNGP of a 2-NN for the cosine (or sine) activation function, we give it here for completeness. The NNGP for $\sigma(x) = \cos(ax)$ equals

$$K_{\cos}(\mathbf{x}, \mathbf{x}') = \mathbb{E}_{\boldsymbol{\omega} \sim \mathcal{N}(\mathbf{0}, I_n)}[\cos(a\boldsymbol{\omega}^T \mathbf{x})\cos(a\boldsymbol{\omega}^T \mathbf{x}')] =$$

$$\mathbb{E}_{\boldsymbol{\omega} \sim \mathcal{N}(\mathbf{0}, I_n)}\left[\frac{1}{4}(e^{ia\boldsymbol{\omega}^T(\mathbf{x}-\mathbf{x}')} + e^{-ia\boldsymbol{\omega}^T(\mathbf{x}-\mathbf{x}')} + e^{ia\boldsymbol{\omega}^T(\mathbf{x}+\mathbf{x}')} + e^{-ia\boldsymbol{\omega}^T(\mathbf{x}+\mathbf{x}')})\right] =$$

$$\frac{1}{2}e^{-\frac{a^2\|\mathbf{x}-\mathbf{x}'\|^2}{2}} + \frac{1}{2}e^{-\frac{a^2\|\mathbf{x}+\mathbf{x}'\|^2}{2}} = e^{-\frac{a^2\|\mathbf{x}\|^2}{2}}e^{-\frac{a^2\|\mathbf{x}'\|^2}{2}}\cosh(a^2\mathbf{x}^\top\mathbf{x}'),$$

and for the sine case equals

$$K_{\sin}(\mathbf{x}, \mathbf{x}') = \mathbb{E}_{\boldsymbol{\omega} \sim \mathcal{N}(\mathbf{0}, I_n)}[\sin(a\boldsymbol{\omega}^T \mathbf{x})\sin(a\boldsymbol{\omega}^T \mathbf{x}')] =$$

$$\mathbb{E}_{\boldsymbol{\omega} \sim \mathcal{N}(\mathbf{0}, I_n)}\left[\frac{1}{4}(e^{ia\boldsymbol{\omega}^T(\mathbf{x}-\mathbf{x}')} + e^{-ia\boldsymbol{\omega}^T(\mathbf{x}-\mathbf{x}')} - e^{ia\boldsymbol{\omega}^T(\mathbf{x}+\mathbf{x}')} - e^{-ia\boldsymbol{\omega}^T(\mathbf{x}+\mathbf{x}')})\right] =$$

$$\frac{1}{2}e^{-\frac{a^2\|\mathbf{x}-\mathbf{x}'\|^2}{2}} - \frac{1}{2}e^{-\frac{a^2\|\mathbf{x}+\mathbf{x}'\|^2}{2}} = e^{-\frac{a^2\|\mathbf{x}\|^2}{2}}e^{-\frac{a^2\|\mathbf{x}'\|^2}{2}}\sinh(a^2\mathbf{x}^\top\mathbf{x}').$$

*Proof of Theorem 8.* For $\mathbf{x}, \mathbf{x}' \in \mathbb{S}^{n-1}$ we have $K_{\cos}(\mathbf{x}, \mathbf{x}') = e^{-a^2}\cosh(a^2\mathbf{x}^\top\mathbf{x}')$ and $K_{\sin}(\mathbf{x}, \mathbf{x}') = e^{-a^2}\sinh(a^2\mathbf{x}^\top\mathbf{x}')$. Let $\lambda_k$ be the eigenvalue of order $k$ of $O_{K_{\cos}}$. The Funk-Hecke formula gives us $\lambda_k \propto_n \int_{-1}^{1} \cosh(a^2t)P_k(t)(1-t^2)^{\frac{n-3}{2}} dt$.

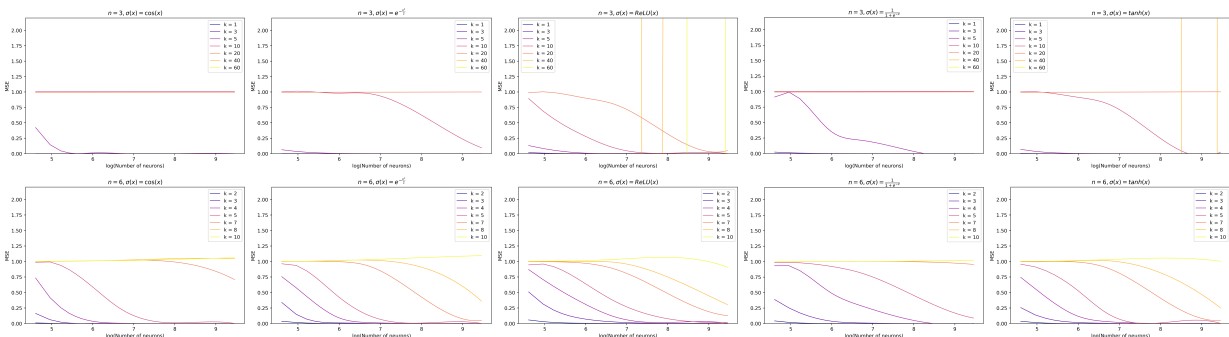

Figure 5: Achieved MSE when learning $Y_k$ by random features model as a function of the number of hidden neurons ($n = 3, 6$).

The latter expression is zero for an odd $k$. For an even $k$, using Rodrigues' formula, we have

$$\lambda_{2k} \propto^n e^{-a^2} \frac{\Gamma(\frac{n-1}{2})}{2^{2k}\Gamma(2k + \frac{n-1}{2})} \int_{-1}^{1} \cosh(a^2 t) \frac{d}{dt^{2k}}[(1-t^2)^{2k + \frac{n-3}{2}}]dt \propto^n$$

$$\frac{a^{4k}e^{-a^2}}{2^{2k}\Gamma(2k + \frac{n-1}{2})} \int_{-1}^{1} \cosh(a^2 t)(1-t^2)^{2k + \frac{n-3}{2}}dt \leq \frac{a^{4k}e^{-a^2}\cosh(a^2)}{2^{2k}\Gamma(2k + \frac{n-1}{2})} \int_{0}^{1} (1-u)^{-\frac{1}{2}}u^{2k + \frac{n-3}{2}}du =$$

$$\frac{a^{4k}\cosh(a^2)e^{-a^2}\text{Beta}(\frac{1}{2}, 2k + \frac{n-1}{2})}{2^{2k}\Gamma(2k + \frac{n-1}{2})} \ll^n \frac{a^{4k}}{\sqrt{k}2^{2k}\Gamma(2k + \frac{n-1}{2})}.$$

Analogously, let $\lambda'_k$ be the eigenvalue of degree $k$ of $O_{K_{\sin}}$. Then we have $\lambda'_k = 0$ for an even $k$ and $\lambda'_{2k+1} \ll^n \frac{a^{4k+2}}{\sqrt{k}2^{2k}\Gamma(2k + \frac{n+1}{2})}$. $\square$

## J OTHER EXPERIMENTS

In Figure 5, results of RFM for dimensions $n = 3, 6$ are given. They only verify the conclusion made in the main part of the paper: the speed of decay of the NNGP kernel's eigenvalues define which activation functions succeed in training $Y_k$ by RFM.

We made experiments with training of a 2-NN after weights were initialized by RFM. The number of hidden neurons was set to 1000, and the optimization was made by Adam with learning rate 0.01. In Figure 6, plots of the MSE dynamics for different activation functions are given. We see the same pattern that was observed by an ordinary training (i.e. without an RFM initialization) — the best performance was demonstrated by the cosine and the Gaussian activation functions. Recall that ReLU outperformed both the cosine and the Gaussian activation function for RFM (see Figure 3). In Section 5 we explained a better performance of ReLU by a less rapid decay of eigenvalues of the corresponding NNGP kernel. This additionally indicates that the structure of the NNGP kernel only reflects properties of the initialization step, and a proper training of weights "forgets" that specifics.

In figure 7, plots the MSE dynamics during training of $Y_k$ by Adam with a standard initialization are given, for $n = 3, 6$. Again, the cosine and the gaussian activations outperform ReLU.

Our computing infrastructure for these experiments is as follows: CPU Intel Core i9-10900X CPU @ 3.70GHz, GPU 2× Nvidia RTX 3090, RAM 128 Gb, Operating System Ubuntu 22.04.1 LTS, torch 1.13.1, cuda 11.7, pandas 1.5.3.

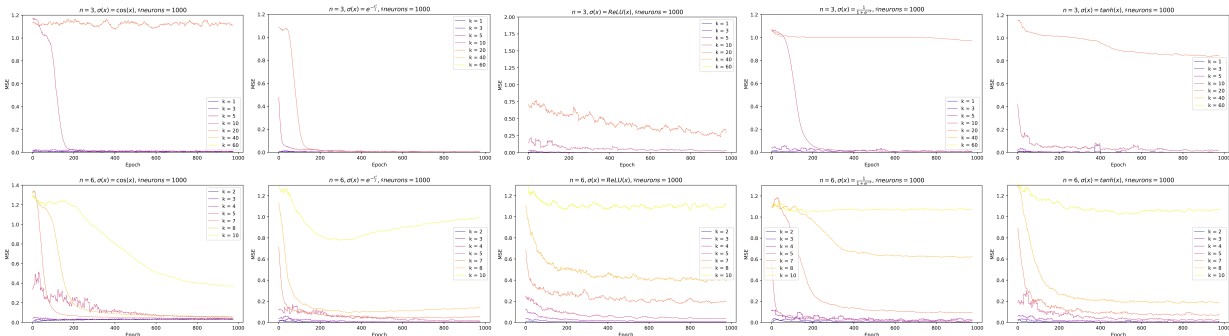

Figure 6: MSE dynamics during learning $Y_k$ with a 2-NN (1000 hidden neurons) after an initialization of weights by RFM for $n = 3, 6$ (rows) and using different activation functions (columns).

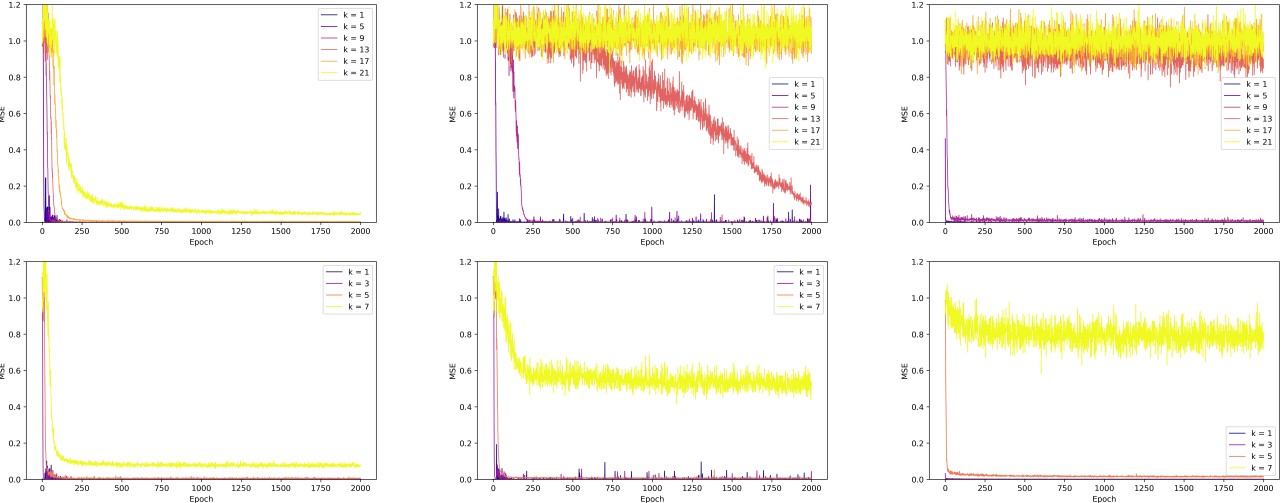

Figure 7: MSE dynamics during learning $Y_k$ with a 2-NN (with a standard initialization) for $n = 3, 6$ (rows) and using activation functions (columns): (a) $\sigma(x) = e^{-\frac{x^2}{2}}$, (b) $\sigma(x) = \cos(x)$, (c) $\sigma(x) = \mathrm{ReLU}(x)$.