# OpenReview forum: "Multi-layer random features and the approximation power of neural networks"
_auai.org/UAI/2024/Conference — UAI 2024 poster_

### Official Review · Reviewer_LV52 · 2024-03-22

**Q2-1 Originality-Novelty:** 3
**Q2-2 Correctness-Technical Quality:** 4
**Q2-5 Clarity Of Writing:** 4

**Q1 Summary And Contributions:**

The paper analyzes Neural architecture under the light of Kernels in the context of Gaussian Processes.
The paper builds a sound and original theoretical analysis for the relationship between NNs and RKHS.
The paper also surfaces discrepancies between the theoretical guarantees and the practical learnability of functions by finite (realistic) networks

**Q2-3 Extent To Which Claims Are Supported By Evidence:**

4: Excellent: all claims are supported by very convincing evidence (in the form of comprehensive experimental evaluation, rigorous mathematical proofs, detailed (pseudo-)code, precise references, well-motivated and realistic assumptions) and the authors deliver what they promise.

**Q2-4 Reproducibility:**

3: Good: key resources (e.g. proofs, code, data) are available and key details (e.g. proofs, experimental setup) are sufficiently well-described for competent researchers to confidently reproduce the main results.

**Q3 Main Strengths:**

* The paper addresses the fundamental topic of analyzing NNs under the light of Kernels
* The quality of analysis and its depth are with no contest worth publishing
* The investigation of discrepancy between theoretical and "realistic" findings is a food for further research and discussions

**Q4 Main Weakness:**

No clear weaknesses. The paper is solid

**Q5 Detailed Comments To The Authors:**

(minor) Please explicitly define $\mathbb{S}^n$ as the n-dim sphere to make the notation self-contained.

**Q9 Complying With Reviewing Instructions:**

Yes

---

> ### Author Rebuttal · Authors · 2024-04-09
>
> Thank you a lot for your review.
>
> Minor: “Please explicitly define S^{n-1} as the n-dim sphere to make the notation self-contained”
>
> We will do that.

---

### Official Review · Reviewer_hHL6 · 2024-03-23

**Q2-1 Originality-Novelty:** 3
**Q2-2 Correctness-Technical Quality:** 3
**Q2-5 Clarity Of Writing:** 3

**Q1 Summary And Contributions:**

This paper studies the approximation power of multi-layer feedforward neural networks by relating it to the RKHS norm defined by the Neural Network Gaussian Process (NNGP) kernel. The key contribution is showing that the unit ball in this RKHS contains only functions that can be well approximated by the corresponding neural network architecture. The number of neurons required for a certain approximation error is determined by the RKHS norm of the target function. The paper also compares this result to the classical Barron's theorem for two-layer networks, characterizing when the RKHS provides a more powerful approximation.

**Q2-3 Extent To Which Claims Are Supported By Evidence:**

3: Good: the main claims are supported by convincing evidence (in the form of adequate experimental evaluation, proofs, (pseudo-)code, references, assumptions).

**Q2-4 Reproducibility:**

3: Good: key resources (e.g. proofs, code, data) are available and key details (e.g. proofs, experimental setup) are sufficiently well-described for competent researchers to confidently reproduce the main results.

**Q3 Main Strengths:**

- Showing that the number of neurons required to achieve a certain approximation error is determined by the RKHS norm of the target function. This quantifies the link between function complexity and network size.
- Comparing the result to Barron's theorem for two-layer networks and characterizing when the RKHS norm provides a better approximation through the decay rate of the NNGP kernel's eigenvalues is an interesting finding.

**Q4 Main Weakness:**

- Reading this paper can be a bit overwhelming. The authors might clean up the body of the paper and move some of the theorems into the appendix for clarity.

**Q5 Detailed Comments To The Authors:**

None.

**Q9 Complying With Reviewing Instructions:**

Yes

---

> ### Author Rebuttal · Authors · 2024-04-09
>
> Thank you for your review. Let us address the question.
>
> “The authors might clean up the body of the paper and move some of the theorems into the appendix for clarity.”
>
> Since all proofs are already in the appendix, we can move statements of Theorem 7 and 8 into the appendix, and give only clarifying remarks instead.

---

### Official Review · Reviewer_Cvke · 2024-03-27

**Q2-1 Originality-Novelty:** 2
**Q2-2 Correctness-Technical Quality:** 3
**Q2-5 Clarity Of Writing:** 2

**Q1 Summary And Contributions:**

This paper focuses on the relationship between neural architectures with randomly initialized weights and Gaussian Random Fields, particularly focusing on the Neural Network Gaussian Process (NNGP) kernel. Here is a concise summary of the main points discussed:

Equivalence to Gaussian Random Fields: The document establishes a link between certain neural network architectures and Gaussian Random Fields, with the NNGP kernel playing a key role in defining the function spaces of these architectures.

Approximation Capabilities: It explores how neural networks with randomly initialized weights can approximate functions, emphasizing the impact of activation functions and the number of neurons in achieving specific approximation errors.

Theoretical Insights: The document presents theoretical results, including Theorem 4, which offer guarantees on function approximation by neural architectures. These results underscore the significance of the NNGP kernel and the boundedness of activation functions in determining the approximation performance of neural networks.

Computational Validation: The PDF likely includes computational experiments or simulations to validate the theoretical findings and demonstrate the practical implications of the results in real-world applications.

In essence, the PDF provides a deep dive into the theoretical foundations of neural network approximation using random features, shedding light on the mathematical principles behind neural architectures with randomly initialized weights.

**Q2-3 Extent To Which Claims Are Supported By Evidence:**

3: Good: the main claims are supported by convincing evidence (in the form of adequate experimental evaluation, proofs, (pseudo-)code, references, assumptions).

**Q2-4 Reproducibility:**

4: Excellent: key resources (e.g. proofs, code, data) are available and key details (e.g. proof sketches, experimental setup) are comprehensively described for competent researchers to confidently and easily reproduce the main results.

**Q3 Main Strengths:**

This paper is easy to follow. The theoretical results are intuitive and look like solid.

The contributions are clear by comparing the approximation capabilities of neural networks based on Barron's theorem and multi-layer features construction, highlighting scenarios where the latter approach offers more succinct neural network approximations.

**Q4 Main Weakness:**

1. The topic may be studied heavily. It is also unknown how this theoretical conclusion can guide the setting of the model in practical applications.


2. Missing appropriate references. For examples,

[Shao-Qun Zhang, Fei Wang, and Feng-Lei Fan. Neural Network Gaussian Processes by Increasing Depth. TNNLS. 2022.]

[Bracale, Daniele, Stefano Favaro, Sandra Fortini, and Stefano Peluchetti. Large-width functional asymptotics for deep Gaussian neural networks. ICLR. 2021.]

**Q5 Detailed Comments To The Authors:**

nothing.

**Q9 Complying With Reviewing Instructions:**

Yes

---

> ### Author Rebuttal · Authors · 2024-04-09
>
> Thank you for your review. Let us address the questions.
>
> “The topic may be studied heavily. It is also unknown how this theoretical conclusion can guide the setting of the model in practical applications."
>
> Indeed, there is some interest in random features models and overall, towards links between kernels and NNs. We believe the “high-level reason” is that RKHSs are objects that were carefully studied in the XX century. So approximating NNs by some sets in RKHSs (and vice versa) can help to understand the approximation power of NNs.
> From the perspective of practical applications, such research can shed light on the following question: how the type of an activation function affects the approximation power of NN architecture? Partially, we answer that question (our experiments on the learnability of Y_k by RFM) in Section 5. Namely, we demonstrate that the decay rate of eigenvalues of the NNGP kernel gives some information about the approximation power of NNs. However, we admit that these are initial studies and our conclusions only hold for NNs trained by RFM.
>
> “Missing appropriate references.”
>
> All listed references are indeed relevant. Thanks for pointing to them. We will cite them properly.

---

### Meta-Review · Area_Chair_p8eh · 2024-04-18

The paper focuses on the link between neural networks and Gaussian processes. It makes a theoretically supported link between the size of the network and complexity of functions that can be approximated by the random feature GP approach. The reviewers all had positive comments on the idea, its presentation and its empirical evaluation. It could be a good addition to the conference and of interest to the wider ML community.